# Optimal experimental design for partially observable pure birth processes

Ali Eshragh[1,2], Matthew P. Skerritt[3]*, Bruno Salvy[4], Thomas McCallum[5]

**1** Carey Business School, Johns Hopkins University, Washington, District of Columbia, United States of America, **2** International Computer Science Institute, University of California at Berkeley, Berkeley, California, United States of America, **3** Discipline of Mathematics, RMIT University, Melbourne, Victoria, Australia, **4** INRIA, Ens de Lyon, Lyon, France, **5** School of Information and Physical Sciences, University of Newcastle, Newcastle, New South Wales, Australia

* matt.skerritt@rmit.edu.au

**Data availability statement:** The data, as well as the code to generate it, are available in a

## Abstract

We develop an efficient algorithm to find optimal observation times by maximizing the Fisher information for the birth rate of a partially observable pure birth process involving $n$ observations. Partially observable implies that at each of the observation time points for counting the number of individuals present in the pure birth process, each individual is observed independently with a fixed probability $p$, modeling detection difficulties or constraints on resources. We apply concepts and techniques from generating functions, using a combination of symbolic and numeric computation, to establish a recursion for evaluating and optimizing the Fisher information. The recursion, while still computationally intensive, greatly improves on previously known computational methods which quickly became intractable even in the $n = 2$ case. Our numerical results reveal the efficacy of this new method. An implementation of the algorithm is available publicly.

## Introduction

*Optimal experimental design* is a statistical methodology for selecting efficient and effective ways to gather data (c.f., [1,2]). It aims to maximize the amount of information obtained from the experiments, quantified by the Fisher information. The significance of Fisher information lies in its connection to the asymptotic variance of maximum likelihood estimators. By leveraging Fisher information one can find (asymptotically) unbiased estimators with minimum-variances, where the optimality of a design depends on the statistical model and is assessed with respect to some criterion. As an example, the widely recognized "D-optimality" criterion focuses on maximizing the determinant of the Fisher information. This motivates the computation of the Fisher information for different statistical and stochastic models, particularly for continuous-time Markov chains.

Continuous-time Markov chains (CTMCs) have gained significant popularity in modeling a range of phenomena such as evolutionary, ecological, and epidemiological processes, owing to their capability to efficiently capture the discrete, interactive, and stochastic aspects of these processes (see, e.g., [3–5]). A crucial element in the application of CTMCs is the estimation of model parameters. Initial research was directed at parameter estimation for

GitHub repository. https://github.com/matt-sk/POPBP-Fisher-Information-Optimisation.git (The data can be found in the 'data' folders underneath the 'Maple/Optimisation/*' folders).

**Funding:** The author(s) received no specific funding for this work.

**Competing interests:** The authors have declared that no competing interests exist.

stochastic processes like pure birth (e.g., [6]), pure death (e.g., [7]), and birth-and-death processes (e.g., [8–12]), under both continuous and equidistant discrete observation intervals. The scope was broadened by Becker & Kersting [13], who were successful in formulating an explicit expression for the Fisher information pertaining to the pure birth process, and derived optimal observation times for a pure birth process by applying optimal experimental design methods.

The applicability of pure birth process was further broadened by Bean, Elliot, Eshragh, & Ross [14] and Bean, Eshragh, & Ross [15]. They investigated the Fisher information for the pure birth process under discrete, non-equidistant observation times, namely: the partially observable pure birth process (POPBP), where each observation is modeled as a binomial random variable dependent on the actual population size. This approach adds realism and complexity to the analysis, particularly relevant in the early stages of pest and disease invasions or cell growth experiments, where smaller populations make diffusion approximations less effective. Bean et al. [14] showed that the POPBP is non-Markovian under any order. In addition, Bean et al. [15] developed an efficient approximation to find and optimize the Fisher information, which was previously restricted to only two observations. As a practical application, Eshragh et al. [16] recently utilized these results to model and analyze the dynamics of the COVID-19 population in its early stages in Australia. Our work in this paper improves upon the foundations and methods of those papers.

The use of generating functions in combinatorics and probability theory is classical (e.g., [17–21]). In many cases, notably in relation to Markov chains, the generating functions turn out to be rational functions, that are themselves related to linear recurrences with constant coefficients. This is the situation we encounter in this article and exploit algorithmically to establish a recursion to compute the Fisher information for a POPBP. This technique allows us to efficiently derive the optimal experimental design numerically for more than two observations. To the best of our knowledge, this is the first attempt in applying generating functions in the context of optimal experimental design.

This article is structured as follows: Optimal experimental design presents optimal experimental design methods and shows how to find the Fisher information for a pure birth process. Partially observable pure birth process introduces the partially observable pure birth process, formulates its Fisher information, and develops analytical results for the structure of optimal experimental design. Generating functions for the likelihood applies generating function techniques to establish an efficient recursion for calculating the Fisher information for partially observable pure birth processes. The 'Experimental methodology' and 'Experimental results' sections exploit the methodology developed in Generating functions for the likelihood to run comprehensive numerical experiments for different values of model parameters. Conclusion concludes and addresses future work.

### Notation

The notation used in this paper is summarized in Table 1 on the following page.

## Optimal experimental design

Consider the stochastic process $\{X_t, t \geq 0\}$, where the random variable $X_t$ is characterized by a probability mass/density function $f_{X_t}(x_t; \boldsymbol{\theta})$, with $\boldsymbol{\theta} = (\theta_1, \dots, \theta_k)$ representing an unknown parameter vector. To estimate this vector accurately, we employ the method of maximum likelihood estimation (MLE). This involves taking $n$ observations $X_{t_1}, \dots, X_{t_n}$ of the process

**Table 1. Table of notation.**

| Symbol | Description |
|---|---|
| $X_t$ | Population size at time $t$ in the pure birth process (PBP) |
| $Y_t$ | Observed population size at time $t$ in the POPBP |
| $\lambda$ | Birth rate parameter of the process |
| $p$ | Probability of observing an individual in the POPBP |
| $n$ | Number of observations |
| $t_1, \dots, t_n$ | Observation times |
| $t_i^*$ | Optimal time for the $i^{th}$ observation |
| $s_i^*$ | Approximate optimal time from asymptotic formula |
| $x_t$ | Realization of $X_t$ |
| $y_t$ | Realization of $Y_t$ |
| $x_0$ | Known initial population size at time $t = 0$ |
| $\boldsymbol{X}_n$ | Random vector of actual population sizes $(X_{t_1}, \dots, X_{t_n})$ |
| $\boldsymbol{Y}_n$ | Random vector of partially observed population sizes $(Y_{t_1}, \dots, Y_{t_n})$ |
| $\boldsymbol{x}_n$ | Realization of actual population sizes $(x_{t_1}, \dots, x_{t_n})$ |
| $\boldsymbol{y}_n$ | Realization of partially observed population sizes $(y_{t_1}, \dots, y_{t_n})$ |
| $\bar{\boldsymbol{y}}_n$ | The vector $(x_0, \boldsymbol{y}_n) = (x_0, y_{t_1}, \dots, y_{t_n})$ |
| $\tau$ | Time horizon of the experiment |
| $\mathcal{L}(\boldsymbol{X}_n \mid \theta)$ | Likelihood function based on observations $\boldsymbol{X}_n$ |
| $\mathcal{FI}_{\boldsymbol{X}_n}(\lambda)$ | Fisher Information from PBP observations |
| $f_{\boldsymbol{X}_n}(\boldsymbol{x}_n; \theta)$ | Joint probability distribution of $\boldsymbol{X}_n$ parameterized by $\theta$ |
| $\mathbb{E}$ | Expected value |
| Var | Variance |
| $\mathcal{FI}_{\boldsymbol{Y}_n}(\lambda)$ | Fisher Information from POPBP observatio |
| $\widehat{\mathcal{FI}}_{\boldsymbol{Y}_2}(\lambda)$ | Approximate Fisher Information for POPBP with $n = 2$ observations |
| $\nu_{i,j}$ | $e^{-\lambda(t_j - t_i)}$ |
| $\vartheta_t$ | $e^{-\lambda t}$ |
| $\eta_i(\boldsymbol{y}_n, \boldsymbol{x}_n)$ | Term in the POPBP likelihood function (c.f., Eq (7)) |
| $Q_i$ | recursive polynomial in $u_0, \dots, u_i$; used in generating functions |
| $\bar{\boldsymbol{u}}^{\bar{c}}$ | Shorthand for $u_0^{c_0} \cdots u_n^{c_n}$ where $\bar{c} = (c_0, \dots, c_n) \in \mathbb{N}^{n+1}$. |
| $q_{\bar{c}}$ | The coefficient of $\bar{\boldsymbol{u}}^{\bar{c}}$ in the polynomial $Q_n$ |
| $q_c$ | The coefficient of $\bar{\boldsymbol{u}}^{(1,c)}$ in the polynomial $Q_n$ for $\boldsymbol{c} = (c_1, \dots, c_n) \in \mathbb{N}^n$ |
| $p_{\bar{\boldsymbol{y}}_n}$ | The coefficient of $\bar{\boldsymbol{u}}^{\bar{\boldsymbol{y}}_n}$ in the numerator of Eq (14) |
| $\mathcal{SL}_{S, \boldsymbol{Y}_n}(\lambda)$ | The $S^{th}$ slice of the Fisher Information computation |
| $\mathfrak{D}_i(\lambda)$ | Drop value of $p$ at which $t_i^*$ drops from 1 |

and maximizing the likelihood function:

$$\mathcal{L}(\boldsymbol{X}_n \mid \theta) = f_{\boldsymbol{X}_n}(\boldsymbol{x}_n; \theta),$$

where $\boldsymbol{X}_n = (X_{t_1}, \dots, X_{t_n})$ denotes a random vector of observations, $\boldsymbol{x}_n = (x_{t_1}, \dots, x_{t_n})$ its realization, and $f_{\boldsymbol{X}_n}(\boldsymbol{x}_n; \theta)$ the joint probability mass/density function of the observed sample.

It is well-known that MLEs asymptotically follow a normal distribution, characterized by a variance matrix denoted as $\text{Var}(\theta)$. The inverse of this matrix introduced in Definition 1 plays a crucial role in statistical estimation theory.

**Definition 1** (Fisher Information). *The inverse of the variance matrix* $\text{Var}(\theta)$*, referred to as the Fisher information matrix and denoted by* $\mathcal{FI}$*, is a $k \times k$ matrix defined as:*

$$\mathcal{FI}(\theta) = \text{Var}(\theta)^{-1}.$$

The Fisher information matrix plays a key role in quantifying the amount of information that a random sample carries about an unknown parameter upon which the likelihood depends [22].

An *optimal experimental design* is defined as an experimental design that optimizes an appropriate function of the Fisher information matrix [2]. Common optimality criteria identified in the literature include:

- *A-optimality*: Minimizing the trace of the inverse Fisher information matrix, which is equivalent to minimizing the trace of the variance matrix,
- *D-optimality*: Maximizing the determinant of the Fisher information matrix,
- *E-optimality*: Maximizing the minimum eigenvalue of the Fisher information matrix,
- *T-optimality*: Maximizing the trace of the Fisher information matrix.

It is important to note that if the parameter vector $\theta$ contains only a single parameter, then $\mathcal{FI}(\theta)$ becomes a scalar. In this simplified scenario, all the above criteria converge, effectively becoming equivalent to maximizing the Fisher information.

The Fisher information matrix can be calculated through one of the two following expectations (see, e.g., [23] Chapter 13):

$$\mathcal{FI}_{X_n}(\theta) = \mathbb{E}_{\mathcal{L}(X_n|\theta)} \left( \nabla_\theta \log(\mathcal{L}(X_n \mid \theta))^T \nabla_\theta \log(\mathcal{L}(X_n \mid \theta)) \right) \tag{1}$$

$$= -\mathbb{E}_{\mathcal{L}(X_n|\theta)} \left( H_\theta \left( \log(\mathcal{L}(X_n \mid \theta)) \right) \right), \tag{2}$$

where $\nabla_\theta$ denotes the gradient vector and $H_\theta$ the Hessian matrix, both with respect to the parameter vector $\theta$. Note that in Eq (1) the superscript $T$ indicates the transpose operation. For a single parameter ($k = 1$), these expressions simplify to the first and second derivatives of the log-likelihood function with respect to $\theta$, thus reducing the Fisher information to a scalar.

Eq (2) demonstrates that the calculation of the Fisher information matrix relies on the likelihood function $\mathcal{L}(X_n \mid \theta)$. Thus, if computing the likelihood function is complex or infeasible, this complexity is likely to carry over to the Fisher information's calculation. Additionally, even possessing an explicit expression for the likelihood function does not guarantee the straightforward derivation of the Fisher information.

An exception to these challenges occurs in the case of observations derived from a *pure birth process*, where both the likelihood function and the Fisher information can be explicitly determined, presented in Definition 2.

**Definition 2** (Pure Birth Process, PBP). *The stochastic process $\{X_t, t \geq 0\}$ is called a pure birth process (PBP) with a birth rate parameter $\lambda > 0$, if $x_t$ represents the population size at time $t$ with the transition rate to the next state, $x_t + 1$, is precisely $\lambda x_t$.*

Throughout, we assume the initial population size, $x_0$, at time $t_0 = 0$ is known. Furthermore, for $0 \leq t_1 < t_2$, the conditional probability mass function of the random variable $(X_{t_2} \mid X_{t_1} = x_{t_1})$ over the values of $x_{t_2} = x_{t_1}, x_{t_1} + 1, \dots$ is

$$P_{(X_{t_2}|X_{t_1})}(x_{t_2} \mid x_{t_1}) = \binom{x_{t_2} - 1}{x_{t_1} - 1} v_{1,2}^{x_{t_1}} (1 - v_{1,2})^{x_{t_2} - x_{t_1}}, \tag{3}$$

where $v_{1,2} = e^{-\lambda(t_2 - t_1)}$ (see, e.g., [24] Chapter 5).

Becker & Kersting [13] extensively studied the Fisher information of observations obtained from the PBP to estimate the unknown birth rate parameter $\lambda$. They demonstrated that for observations $X_{t_1}, \dots, X_{t_n}$ from the PBP with parameter $\lambda$ at specific observation times $t_1, \dots, t_n$

within a predetermined time horizon $t_n = \tau$, the likelihood function for these observations is given by:

$$\mathcal{L}_{X_n}(x_n \mid \lambda) = \prod_{i=1}^{n} \binom{x_{t_i} - 1}{x_{t_{i-1}} - 1} v_{i-1,i}^{x_{t_i}} (1 - v_{i-1,i})^{x_{t_i} - x_{t_{i-1}}}, \qquad (4)$$

where $v_{i-1,i} = e^{-\lambda(t_i - t_{i-1})}$. This representation of the likelihood function, as a product, facilitates the evaluation of the Fisher information via Eq (2).

Utilizing this formulation, Becker & Kersting [13] derived an explicit expression for the Fisher information for the random observation vector $X_n$ used in estimating $\lambda$:

$$\mathcal{FI}_{X_n}(\lambda) = x_0 \sum_{i=1}^{n} \frac{(t_i - t_{i-1})^2}{e^{-\lambda t_{i-1}} - e^{-\lambda t_i}}. \qquad (5)$$

Furthermore, they showed that with given values of $\tau$, $n$, and $\lambda$, the optimal experimental design $(t_1^*, t_2^*, \ldots, t_n^*)$ can be uniquely determined by solving the following optimization equations:

$$\varphi_1(\lambda(t_i - t_{i-1})) = \varphi_2(\lambda(t_{i+1} - t_i)) \ \text{ for } i = 1, \ldots, n-1,$$

where the functions $\varphi_1$ and $\varphi_2$ are defined as:

$$\varphi_1(x) := \frac{x(2e^x - x - 2)}{(e^x - 1)^2}, \quad \varphi_2(x) := \frac{xe^x(2e^x - xe^x - 2)}{(e^x - 1)^2}.$$

For large sample sizes, an approximate solution to these equations simplifies the experimental design process:

$$t_i^* \approx s_i^* := \frac{3}{\lambda} \log\left(1 + \frac{i}{n}\left(e^{\frac{\lambda \tau}{3}} - 1\right)\right) \ \text{ for } i = 1, \ldots, n. \qquad (6)$$

We compare these approximate $s_i^*$ against $t_i^*$ calculated directly (by optimizing Eq (5)) in the Optimization method of the Experimental methodology, below.

Although this approach is interesting, it may not be practical due to real-world restrictions, such as time and budget constraints, which may prevent us from observing and counting all individuals in the population at each observation time $t_i$. To address this issue, we employ a modified stochastic process, a *partially observable pure birth process* (POPBP), which will be explained in Partially observable pure birth process.

## Partially observable pure birth process

Consider a PBP $\{X_t, t \geq 0\}$ with an unknown birth rate $\lambda$. To estimate this unknown parameter $\lambda$, we aim to take $n$ observations at times $0 \leq t_1 \leq \cdots \leq t_n = \tau$. Let us now assume that at each observation time $t_i$, we may not be able to observe the entire population size $X_{t_i}$ but can only observe a random sample from it. Consequently, we define a POPBP as follows:

**Definition 3** (Partially Observable Pure Birth Process, POPBP [14]). *Consider the* PBP *$\{X_t, t \geq 0\}$ with birth rate $\lambda$. If random variables $Y_t$ is defined such that the conditional random variable $(Y_t \mid X_t = x_t)$ follows the* $\text{Bin}(x_t, p)$ *distribution, where*

$$P_{(Y_t|X_t)}(y_t \mid x_t) = \binom{x_t}{y_t} p^{y_t} (1-p)^{x_t-y_t} \quad \textit{for } y_t = 0, \ldots, x_t,$$

*then the stochastic process $\{Y_t, t \geq 0\}$ is called the partially observable pure birth process (POPBP) with parameters $(\lambda, p)$.*

**Remark 4.** *Definition 3 implies that, for a population size at time $t \in (0, \infty)$ equal to $x_t$, where each of these $x_t$ individuals can be observed independently with probability $p$, the random variable $Y_t$ then counts the total number of observed individuals at that time. Consequently, the POPBP with parameters $(\lambda, 1)$ simplifies to the PBP with the same parameter $\lambda$, because when $p = 1$, every individual in the population is observed, mirroring the observation conditions of a PBP. Furthermore, it is assumed throughout that the parameter $p$ is both fixed and known.*

Bean et al. [14] demonstrated that the POPBP $\{Y_t, t \geq 0\}$, characterized by parameters $(\lambda, p)$, does not exhibit the Markovian property. This characteristic means that the likelihood function for observations from the POPBP cannot be simplified utilizing the Markovian property, in contrast to the PBP. Taking into account this significant difference, Bean et al. [15] derived the likelihood function for the POPBP as follows:

$$\mathcal{L}_{Y_n}(\boldsymbol{y}_n \mid \lambda, p) = \sum_{x_0 \leq x_{t_1} \leq \cdots \leq x_{t_n}} \prod_{i=1}^{n} \eta_i(\boldsymbol{y}_n, \boldsymbol{x}_n), \tag{7}$$

where

$$\eta_i(\boldsymbol{y}_n, \boldsymbol{x}_n) = \binom{x_{t_i}}{y_{t_i}} p^{y_{t_i}} (1-p)^{x_{t_i}-y_{t_i}} \binom{x_{t_i}-1}{x_{t_{i-1}}-1} v_{i-1,i}^{x_{t_{i-1}}} (1-v_{i-1,i})^{x_{t_i}-x_{t_{i-1}}}.$$

Eq (7) reveals that, unlike the likelihood function for a PBP (i.e., Eq (4)), the likelihood function for the POPBP cannot be represented simply as a product form. This complexity suggests that calculations involving the likelihood function, including those for the Fisher information, will be considerably more challenging than those for the PBP.

The Fisher information for the parameter $\lambda$, based on $n$ observations of the POPBP, is presented as:

$$\mathcal{FI}_{Y_n}(\lambda) = \sum_{0 \leq \boldsymbol{y}_n} \frac{(\frac{\partial}{\partial \lambda} \mathcal{L}_{Y_n}(\boldsymbol{y}_n \mid \lambda, p))^2}{\mathcal{L}_{Y_n}(\boldsymbol{y}_n \mid \lambda, p)}. \tag{8}$$

The calculation of the partial derivative in Eq (8) can be stated in terms of functions $\eta_i$ as follows:

$$\frac{\partial}{\partial \lambda} \mathcal{L}_{Y_n}(\boldsymbol{y}_n \mid \lambda, p) = \sum_{1 \leq x_{t_1} \leq \cdots \leq x_{t_n}} \sum_{j=1}^{n} (t_j - t_{j-1}) \left( \frac{x_{t_j} v_{j-1,j} - x_{t_{j-1}}}{1 - v_{j-1,j}} \right) \prod_{i=1}^{n} \eta_i(\boldsymbol{y}_n, \boldsymbol{x}_n). \tag{9}$$

Substituting Eqs (7), (9) into Eq (8) allows for the calculation of the Fisher information for the POPBP. However, this process does not lead to a simplified form as seen with the PBP, complicating numerical calculations and optimization efforts.

Notably, the computation in Eq (8) involves $n + 2$ infinite series, including those over $x_{t_n}$ in both the numerator and denominator of the summand, as well as $n$ series over $y_{t_1}, y_{t_2}, \ldots, y_{t_n}$. To achieve a desirable precision level in numerical calculations of the Fisher information,

Bean et al. [15] recommended a *truncation criterion* based on Chebyshev's inequality, coupled with a *relative-error criterion*. This approach ensures that the ratio of the summand to the cumulative sum up to the current point is below a predetermined significance level.

Numerical calculations for a `POPBP` are challenging due to the infinite summations required to compute the Fisher information. This complexity is magnified as $n$, the number of observation times, increases. Specifically, each additional observation necessitates the truncation of three more infinite series: one for calculating $\mathcal{FI}_{Y_n}$ over $y_n$, another for $\mathcal{L}_{Y_n}$ over $x_n$, and a third for the partial derivative $\partial \mathcal{L}_{Y_n}/\partial \lambda$ over $x_n$. Consequently, computation times can become prohibitively long, even for relatively small $n$. For example, with $n = 3$ and $\lambda = 2$, estimating optimal observation times for $p$ ranging from 0.01 to 0.99 in increments of 0.01 is projected to take five years (This estimate is based on calculations implemented in `C++`.), highlighting the significant computational demands.

Moreover, as $\lambda$ increases, computation time escalates exponentially due to the truncation points being exponential functions of $\lambda$. For instance, maximizing the Fisher information for $n = 2$ and varying $p$ from 0.01 to 0.99 by 0.01 steps takes approximately 14 hours for $\lambda = 2$. However, increasing $\lambda$ from 2 to 5 raises the estimated computation time to two years, underscoring the exponential growth in computational demand with parameter increases.

Bean et al. [15] developed an approximation for the Fisher information in the `POPBP` with two observations ($n = 2$) as follows:

$$
\widetilde{\mathcal{FI}}_{Y_2}(\lambda) = \left(1 + \frac{p}{\vartheta_{t_1}}\right)
$$

$$
\times \frac{p\left(p + (1-p)(p\upsilon_{1,2} + (1-p)\vartheta_{t_2}) - (1-p)(p\upsilon_{1,2} + (1-p)\vartheta_{t_2})^2\right)\left((t_2 - t_1)p + (1-p)t_2\vartheta_{t_1}\right)^2}{(p + (1-p)\vartheta_{t_1})^2\left(p + p(1-p)\upsilon_{1,2} + (1-p)^2\vartheta_{t_2}\right)^2\left(1 - p\upsilon_{1,2} - (1-p)\vartheta_{t_2}\right)}
$$

$$
- \left(\frac{p}{p + (1-p)\vartheta_{t_1}}\right)\left(\frac{p(t_2 - t_1)^2(p + (1-p)(1 - \upsilon_{1,2})\upsilon_{1,2})}{(1 - \upsilon_{1,2})(p + (1-p)\upsilon_{1,2})^2}\right) + \frac{pt_1^2(p + (1-p)(1 - \vartheta_{t_1})\vartheta_{t_1})}{(1 - \vartheta_{t_1})(p + (1-p)\vartheta_{t_1})^2}
$$

(10)

where $\vartheta_t := e^{-\lambda t} = \upsilon_{0,t}$. Their work demonstrates, both theoretically and numerically, that Eq (10) provides a highly accurate approximation of $\mathcal{FI}_{Y_2}(\lambda)$. Furthermore, they proved that as $\lambda$ increases, the approximation error quickly diminishes to zero. Significantly, because $\widetilde{\mathcal{FI}}_{Y_2}(\lambda)$ does not involve any infinite summation, it enables the rapid approximation of the Fisher information for any $\lambda$ value.

Unfortunately, while $\widetilde{\mathcal{FI}}_{Y_2}(\lambda)$ offers an excellent approximation for $n = 2$ observation times, extending this approach to higher values of $n$ becomes intractable due to the increasing computational complexity and the absence of straightforward analytical solutions. In Generating functions for the likelihood, we introduce a novel numerical algorithm designed to compute and maximize the Fisher information for the `POPBP` more efficiently, addressing these challenges.

We conclude this section by demonstrating the *rescaling* property of the optimal experimental design for the `POPBP`, as articulated in Definition 5, which plays a crucial role in enhancing the efficiency and applicability of experimental designs.

**Proposition 5.** *If $(t_1^*, \ldots, t_n^*)$ constitutes an optimal experimental design for a* `POPBP` *with parameters $(\lambda, p)$ and a time-horizon of 1, then for any fixed $\tau > 0$, the scaled design $(\tau t_1^*, \ldots, \tau t_n^*)$ forms the corresponding optimal experimental design for a* `POPBP` *with parameters $(\lambda/\tau, p)$ and a time-horizon of $\tau$.*

*Proof*: Denote by $\mathcal{FI}^1$ and $\mathcal{FI}^\tau$ the Fisher information for a POPBP with parameters $(\lambda, p)$ over a time-horizon of 1, and for a POPBP with parameters $(\lambda/\tau, p)$ over a time-horizon of $\tau$, respectively. According to Eq (8), the Fisher information $\mathcal{FI}^1$ for any arbitrary set of sampling times $(t_1, \dots, t_n)$ equates to $\tau^2 \mathcal{FI}^\tau$ for the scaled set $(\tau t_1, \dots, \tau t_n)$. Therefore, if $(t_1^*, \dots, t_n^*)$ maximizes $\mathcal{FI}^1$, the scaled set $(\tau t_1^*, \dots, \tau t_n^*)$ naturally maximizes $\mathcal{FI}^\tau$, establishing its optimality for the latter process. □

**Remark 6.** *Definition 5 implies that to find an optimal experimental design for a given* POPBP *with time-horizon $\tau$, we can find the corresponding optimal experimental design for the rescaled* POPBP *with time-horizon 1 and then, by a simple linear transformation, convert it to an optimal experimental design of the original process. Thus, without loss of generality, we assume henceforth that $\tau = 1$.*

## Generating functions for the likelihood

In this section, which can be considered the main contribution of this paper, we develop a new approach involving the use of generating functions to compute the Fisher information for higher values of $n$ and $\lambda$.

The *generating function* of a sequence $g(z_1, \dots, z_n)$ indexed by non-negative integer variables $z_i$ is the formal power series

$$\phi(u_1, \dots, u_n) = \sum_{z_n=0}^{\infty} \cdots \sum_{z_1=0}^{\infty} g(z_1, \dots, z_n) u_1^{z_1} \cdots u_n^{z_n}.$$

When the generating function $\phi$ is a rational function

$$\phi(u_1, \dots, u_n) = \frac{P(u_1, \dots, u_n)}{Q(u_1, \dots, u_n)}, \tag{11}$$

with two polynomials $P$ and $Q$, the sequence $g(z_1, \dots, z_n)$ satisfies a linear recurrence with constant coefficients obtained by equating the coefficients of the same powers of $u_1^{z_1} \cdots u_n^{z_n}$ on both sides of the identity

$$Q(u_1, \dots, u_n) \sum_{z_n=0}^{\infty} \cdots \sum_{z_1=0}^{\infty} g(z_1, \dots, z_n) u_1^{z_1} \cdots u_n^{z_n} = P(u_1, \dots, u_n). \tag{12}$$

Motivated by this idea, we develop a recursive equation for the likelihood function of a POPBP, which we utilize to calculate and maximize the Fisher information. In our application, the initial population size $x_0$ is known, so it is not an input variable of the likelihood function. There is no difficulty in considering the initial population size as a random variable, which will be constant in our special case. Thus, the generating function of the likelihood function (Eq (7)) we consider is defined by

$$\phi(\boldsymbol{u}_n) = \sum_{y_{t_n}=0}^{\infty} \cdots \sum_{y_{t_1}=0}^{\infty} \sum_{x_0=1}^{\infty} \mathcal{L}_{\boldsymbol{Y}_n}(\boldsymbol{y}_n \mid \lambda, p) u_0^{x_0} \prod_{i=1}^{n} u_i^{y_{t_i}}$$

$$= \sum_{y_{t_n}=0}^{\infty} \cdots \sum_{y_{t_1}=0}^{\infty} \sum_{x_0=1}^{\infty} \sum_{x_{t_n}=\max\{x_0, y_n\}}^{\infty} \cdots \sum_{x_{t_1}=\max\{x_0, y_1\}}^{x_2} u_0^{x_0} \prod_{i=1}^{n} \eta_i(\boldsymbol{y}_n, \boldsymbol{x}_n) u_i^{y_{t_i}}, \tag{13}$$

where $\boldsymbol{u}_n := (u_0, \dots, u_n)$. It turns out that this is a simple rational function.

**Lemma 7.** *Consider the* `POPBP` *with parameters* $(\lambda, p)$ *with* $n \geq 1$ *observations. The generating function of the likelihood function is given by*

$$\phi(\boldsymbol{u}_n) = \frac{u_0 v_{0,1} \cdots v_{n-1,n}(pu_1 + q) \cdots (pu_n + q)}{1 - Q_n}, \tag{14}$$

*where* $q = 1 - p$ *and* $(Q_i)$ *is a family of polynomials defined by* $Q_0 = (1 - v_{0,1}) + v_{0,1} u_0$ *and*

$$Q_i = (1 - v_{i,i+1}) + v_{i,i+1}(pu_i + q)Q_{i-1}, \qquad i \geq 0 \tag{15}$$

*with the convention* $v_{n,n+1} = 1$.

*Proof*: By adopting the convention that $\binom{n}{k}$ is 0 for $k < 0$ and $k > n$, all sums in Eq (13) can be taken over $\mathbb{N}$ (for the variables $y_{t_j}$) or $\mathbb{N} \setminus \{0\}$ (for the variables $x_{t_j}$).

Summation over $y_{t_i}$ using the binomial theorem reduces the generating function to

$$\sum_{x_0 \geq 1, x_{t_1} \geq 1, \dots, x_{t_n} \geq 1} (pu_n + q)^{x_{t_n}} \prod_{i=0}^{n-1} \tilde{\eta}_i,$$

with

$$\tilde{\eta}_i = \binom{x_{t_{i+1}} - 1}{x_{t_i} - 1} w_i^{x_{t_i}} v_{i,i+1}^{x_{t_i}} (1 - v_{i,i+1})^{x_{t_{i+1}} - x_{t_i}}, \quad i \geq 0,$$

where $w_0 = u_0$ and $w_i = pu_i + q$ for $i \geq 1$. By the binomial theorem,

$$\sum_{x_0 \geq 1} \tilde{\eta}_0 = u_0 Q_0^{x_{t_1} - 1}$$

and then by induction

$$\sum_{x_{t_i} \geq 1} \tilde{\eta}_i Q_{i-1}^{x_{t_i} - 1} = v_{i,i+1}(pu_i + q) Q_i^{x_{t_{i+1}} - 1}, \quad i = 1, \dots, n-1.$$

The final sum is

$$\sum_{x_{t_n} \geq 1} (pu_n + q)^{x_{t_n}} Q_{n-1}^{x_{t_n} - 1} = \frac{pu_n + q}{1 - Q_n},$$

which concludes the proof. $\qquad\square$

**Remark 8.** *The important special case when* $x_0$ *is fixed and equal to 1 corresponds to extracting the coefficient of* $u_0^1$ *in this rational function. This is achieved by setting* $u_0 = 1$ *(hence* $Q_0 = 1$*) in* Eq (14).

For notational convenience, we write $\bar{\boldsymbol{y}}_n$ for $(x_0, \boldsymbol{y}_n)$ and if $\bar{\boldsymbol{c}} = (c_0, c_1, \dots, c_n) \in \mathbb{N}^{n+1}$,

$$\mathcal{L}_{Y_n}(\bar{\boldsymbol{y}}_n - \bar{\boldsymbol{c}} \mid \lambda, p) := \mathcal{L}_{Y_n}(x_0 - c_0, y_{t_1} - c_1, \dots, y_{t_n} - c_n \mid \lambda, p),$$

with the convention that this value is 0 if any of the entries of $\bar{\boldsymbol{y}}_n - \bar{\boldsymbol{c}}$ is negative. We use the same notation with $\boldsymbol{y}_n$ and $\boldsymbol{c}$ in the case when $x_0$ is fixed and equal to 1. Also $\bar{\boldsymbol{u}}^{\bar{\boldsymbol{c}}}$ denotes the monomial $u_0^{c_0} \cdots u_n^{c_n}$.

With this notation, a consequence of the explicit form of the generating function is a simple recursion for the likelihood function.

**Theorem 9.** *Consider the* POPBP *with parameters* $(\lambda, p)$ *with* $n \geq 1$ *observations. The likelihood function satisfies the following recurrence equation:*

$$\mathcal{L}_{Y_n}(\bar{\mathbf{y}}_n \mid \lambda, p) = \frac{1}{q_0}\left( p_{\bar{\mathbf{y}}_n} + \sum_{\substack{\bar{c} \in \{0,1\}^{n+1} \\ \bar{c} \neq (0,\ldots,0)}} q_{\bar{c}}\, \mathcal{L}_{Y_n}(\bar{\mathbf{y}}_n - \bar{c} \mid \lambda, p) \right), \tag{16}$$

*where* $q_{\bar{c}}$ *is the coefficient of* $\bar{\mathbf{u}}^{\bar{c}}$ *in the polynomial* $Q_n$ *of* Eq (7)*, while* $p_{\bar{\mathbf{y}}_n}$ *is the coefficient of* $\bar{\mathbf{u}}^{\bar{\mathbf{y}}_n}$ *in the numerator of* Eq (14)*. In the special case when* $x_0$ *is fixed and equal to 1, the same result holds with* $\mathbf{y}_n$ *and* $c$ *in the place of* $\bar{\mathbf{y}}_n$ *and* $\bar{c}$*.*

*Proof*: This is the result of multiplying both sides of Eq (14) by $1 - Q_n$ and extracting the coefficient of $\bar{\mathbf{u}}^{\bar{c}}$ on both sides. □

**Remark 10.** *The coefficients of* $Q_n$ *are easily computed from the recurrence* Eq (15)*. Similarly, the coefficients of the numerator of* Eq (14) *follow easily from its expression. Note that both polynomials have degree 1 in each of the variables* $u_0, \ldots, u_n$*.*

**Remark 11.** *The recurrence* Eq (15) *shows that for* $p$ *and the* $v_{i,i+1}$ *in the interval [0,1], all the coefficients of the recurrence are positive, making it numerically stable. Moreover, if* $p \notin \{0,1\}$ *and all* $v_{i,i+1} \neq 0$ *($i \leq n-1$), all these coefficients are non-zero: the recurrence has exactly* $2^{n+1}$ *terms.*

**Remark 12.** *The formula* Eq (16) *also gives the initial conditions. For instance, with* $\bar{\mathbf{y}}_n = \mathbf{0}$*, it gives* $\mathcal{L}_{Y_n}(\bar{\mathbf{0}} \mid \lambda, p) = p_{\bar{\mathbf{0}}}/q_0$*.*

By taking a derivative with respect to $\lambda$ from both sides of the recurrence equation for the likelihood function given in Definition 9, one obtains a similar recurrence equation for the derivative of the likelihood function. This expression involves the derivative of the coefficients $q_{\bar{c}}$ with respect to $\lambda$. They are the coefficients of the polynomial $\partial Q_n/\partial \lambda$. These polynomials are computed thanks to the recurrence

$$\frac{\partial Q_i}{\partial \lambda} = \frac{\partial v_{i,i+1}}{\partial \lambda}\left((pu_i + q)Q_{i-1} - 1\right) + v_{i,i+1}(pu_i + q)\frac{\partial Q_{i-1}}{\partial \lambda},$$

which can be simplified using $v_{i,i+1} = \exp(-\lambda(t_{i+1} - t_i))$.

By exploiting all these results together, we can calculate the Fisher information for the POPBP using Eq (8) for a given initial population size $x_0$. Note that in numerical evaluations, all infinite sums in the calculation of the Fisher information should be properly truncated.

## Experimental methodology

In Generating functions for the likelihood we used generating functions to develop a recursive equation for the likelihood function of a POPBP. As stated in Partially observable pure birth process, computing and maximizing the Fisher information for a POPBP even for small values of $n$ and $\lambda$ can be very time consuming. Nonetheless, this section shows the results of Generating functions for the likelihood can significantly speed up the computation of the

Fisher information and accordingly help us derive optimal experimental designs for `POPBPs` efficiently. Recall that the goal is to compute the following optimal observation times:

$$(t_1^*, \ldots, t_n^*) \in \mathrm{argmax}\{\mathcal{FI}_{Y_n}(\lambda)\}.$$

We used *Maple 2017* to symbolically pre-compute the generating function for the likelihood function $\mathcal{L}_{Y_n}(y_n; x_0 \mid \lambda, p)$ and its derivative, which are used to compute the Fisher information.

## Parallelization

For a vector $\boldsymbol{c} = (c_0, \ldots, c_n) \in \mathbb{N}^{n+1}$, we call $|\boldsymbol{c}| := \sum_{k=0}^{n} c_k$ its *degree*. A consequence of the recurrence relation for $\mathcal{L}_{Y_n}(y_n; x_0 \mid \lambda, p)$ is that the recursive computation of $\mathcal{L}_{Y_n}(y_n \mid \lambda, p)$ relies entirely on values of $\mathcal{L}_{Y_n}(z_n \mid \lambda, p)$ for vectors $z_n$ of smaller degree (and similarly for $\frac{\partial}{\partial \lambda} \mathcal{L}_{Y_n}(y_n \mid \lambda, p)$). We exploit this observation to enable parallelization of the computation.

**Definition 13** (Slice)**.** *Let S>0 be an integer. We define*

$$\mathcal{SL}_{S,Y_n}(\lambda) := \sum_{|y_n|=S} \frac{\left(\frac{\partial}{\partial \lambda} \mathcal{L}_{Y_n}(y_n \mid \lambda, p)\right)^2}{\mathcal{L}_{Y_n}(y_n \mid \lambda, p)},$$

*and call it the $S^{th}$ slice of the computation of $\mathcal{FI}_{Y_n}(\lambda)$. We write $WC_n(S)$ for the set $\{y_n \mid |y_n| = S\}$.*

Thus the Fisher information may be rewritten

$$\mathcal{FI}_{Y_n}(\lambda) = \sum_{y_{t_1}=0}^{\infty} \cdots \sum_{y_{t_n}=0}^{\infty} \frac{\left(\frac{\partial}{\partial \lambda} \mathcal{L}_{Y_n}(y_n \mid \lambda, p)\right)^2}{\mathcal{L}_{Y_n}(y_n \mid \lambda, p)} = \sum_{S=0}^{\infty} \sum_{|y_n|=S} \frac{\left(\frac{\partial}{\partial \lambda} \mathcal{L}_{Y_n}(y_n \mid \lambda, p)\right)^2}{\mathcal{L}_{Y_n}(y_n \mid \lambda, p)}$$

$$= \sum_{S=0}^{\infty} \mathcal{SL}_{S,Y_n}(\lambda).$$

We compute $\mathcal{FI}_{Y_n}(\lambda)$ by computing each slice in turn, starting at 0, and with the computations for each slice being independently computed in parallel. We store the values of $\mathcal{L}$ and $\partial \mathcal{L}/\partial \lambda$ from the terms of each slice until they are no longer needed.

## Implementation considerations

The above computation method for $\mathcal{FI}_{Y_n}(\lambda)$ was implemented in `C++`, and compiled to a shared library to facilitate its use with third party optimization software. Interested readers can access the code through Github (https://github.com/matt-sk/POPBP-Fisher-Information-Calculator.git).

This implementation includes code written to be multi-threaded so as to compute the terms in an individual slice in parallel. Note that we also wrote and implemented single-threaded code, and both single-threaded and multi-threaded options are included in the implementation. Thus the user may choose at runtime whether to take advantage of multiple processors.

The computation proceeds one slice at a time starting at $S = 0$ and continuing until the sum of values for a slice does not change the accumulated value. That is, we terminate computation when $\sum_{S=0}^{M} \mathcal{SL}_{S,Y_n}(\lambda) = \sum_{S=0}^{M+1} \mathcal{SL}_{S,Y_n}(\lambda)$.

We observe from Eq (16) in Definition 9 that the computation of a single slice needs only the $n$ previous slices. Consequently, our computation only stores the current slice and the $n$ previous slices, discarding no-longer-needed slices as we compute.

Currently, the values of $n$ for which our implementation can compute $\mathcal{FI}_{Y_n}(\lambda)$ are fixed due to the generating functions having been pre-computed for fixed $n$. The coefficients of the recurrence relation from Eq (16) are hard-coded in the software, although in such a way that little modification is needed to add new cases. As of the time of writing, our implementation is capable of computing $\mathcal{FI}_{Y_n}(\lambda)$ for $n \in \{2, 3, 4, 5\}$.

We have written our implementation using C++ templates in such a way that it should be capable of computation using any desired precision. For the templated code to work with arbitrary precision numeric types, those numeric types must use overloaded arithmetic operators. We have not tested it using arbitrary precision libraries; we have used only IEEE single (32 bit) and double (64 bit) precision (C++ `float` and `double` types, respectively). We computed the results in this article with double precision.

Our implementation takes $t_1, \dots, t_n$, $p$, and $\lambda$, and computes the value of $\mathcal{FI}_{Y_n}(\lambda)$. The coefficients $q_c$ and $p_c$ depend on the values of $t_1, \dots, t_n$, so our implementation begins by computing and storing these coefficients. More precisely, our implementation computes and stores $p_{\bar{y}_n}/q_0$ and $q_c/q_0$ so as to save a division operation in the computation of each $\mathcal{L}_{Y_n}(y_n \mid \lambda, p)$ (and thus save many divisions over the computation of $\mathcal{FI}_{Y_n}(\lambda)$).

Each slice is stored in an array. To compute the value of $\mathcal{L}_{Y_n}(y_n \mid \lambda, p)$ for a given $y_n$ we must be able to access arbitrary values within the earlier slices. To do so we must be able to index each element in the slice. That is, we must be able to describe *in code* a bijection between the integers $\{0, \dots, |WC_n(S)| - 1\}$ and the elements of $WC_n(S)$. We have described the bijection with a recurrence relation, and computed it symbolically in *Maple* for the values of $n \in \{1, 2, 3, 4, 5\}$.

We would like a more generic solution to this indexing problem (i.e., one that does not require hard coding for each new value of $n$ for which we want to compute). Such a solution would need to be at least as fast in implementation as our current method. We note that an early iteration of the implementation used an associative container (`std::unordered_map` in C++; readers familiar with Python can think of this as a dictionary) as a generic solution, but this solution proved to be slower than the current implementation, presumably due to the search time inherent in the data structure.

Being able to preserve locality between elements of a slice and the required elements of the lower slices needed for their computation—whether through indexing, a clever data structure, or otherwise—would be particularly desirable. That is, we would like to be able to reliably partition $WC_n(S)$ (roughly evenly) in such a way that each partitioned subset, as well as the subsets of $WC_n(S - 1), \dots, WC_n(S - n)$ required for its computation, are easily extractable in contiguous memory with few unneeded extra elements. Doing so would allow us to more easily break up the computation of a single slice over multiple computation devices (e.g., using GPUs or MPI) and—if the partitions were small enough—could also allow some more fine-grained memory caching optimizations on a single machine.

Finally, we note that although our implementation can compute for the case $n = 5$, no results are printed in this paper for this case. The optimization times for this case proved to be prohibitively slow.

## Optimization method

To optimize $\mathcal{FI}_{Y_n}(\lambda)$ (i.e., find $(t_1^*, t_2^*, \dots, t_n^*) \in \text{argmax}\{\mathcal{FI}_{Y_n}(\lambda)\}$) for fixed values of $\lambda$ and $p$ we use *Maple*'s `NLPSolve` function (that itself relies on numerical code from the NAG

library) to perform the optimization using our `C++` implementation (accessed through the shared library) for computation of the Fisher information.

We know that $t_1^* \leq t_2^* \cdots \leq t_n^*$ and that $t_n^* = 1$. Consequently, we search over the domain $0 \leq t_1 \leq \cdots \leq t_n = 1$. The boundaries of this domain are when $t_1 = \cdots = t_i = 0$ for any $1 \leq i < n$, when $t_i = t_{i+1} = \ldots = t_n = 1$ for any $1 \leq i \leq n$, and when $0 \neq t_i = t_{i+1} = \cdots = t_j \neq 1$ for some $1 \leq i \leq j \leq n$. Note that when $n$ is large enough we may have boundaries that are unions of these types of boundary.

We optimize the interior and each boundary individually. However, we also know—because we know the population size at time $t = 0$ (i.e., $x_0$) almost surely—that $t_i^* \neq 0$ for any $i$, so we exclude any boundaries where $t_i^* = 0$.

The `NLPSolve` function offers different optimization methods, and we use two of them depending on whether the region we are optimizing over is one-dimensional, or multi-dimensional. For one-dimensional regions (boundaries with only a single varying $t_i$, or with a single parameter $t$ for a single group of equal arguments $t_i = \cdots = t_{i+k} = t$) we use the "`branchandbound`" method, which performs a global search. For multi-dimensional regions (all other regions) we use the default method, which for our problem is the "`SQP`" (sequential quadratic programming) method.

**Definition 14.** *Numerical results reveal that for a fixed set of parameters, the value of $t_i^*$ is equal to 1 for small values of $p$. However, a value of $p$ exists for which $t_i^*$ drops suddenly from $t_i^* = 1$. We call such a point a "drop value" and denote it by $\mathfrak{D}_i(\lambda)$. Clearly, $\mathfrak{D}_n(\lambda)$ does not exist because $t_n^*$ is always 1.*

The graphs of $t_i^*$ in Experimental results are produced by first choosing a fixed $\lambda$ and then calculating the so-called *drop values* using a binary search. For every $1 \leq i < n$ we presume $t_i^* = 1$ when $p = 0$, and that $t_i^* \neq 1$ when $p = 1$ and conduct a binary search on $p$ to bound $\mathfrak{D}_i(\lambda)$. We search for $\mathfrak{D}_1(\lambda)$ first, then $\mathfrak{D}_2(\lambda)$, and so on. However, because each optimization produces $t_1^*, \ldots, t_{n-1}^*$ for a particular $p$ value, we update the upper and lower bounds of all $\mathfrak{D}_i(\lambda)$ while we are searching for a particular one. This approach allows the later binary searches to perhaps have narrower bounds to begin searching within.

The bounds for each $\mathfrak{D}_i(\lambda)$ yield an open interval, $0 \leq a_i < \mathfrak{D}_i(\lambda) < b_i \leq 1$, which is stored after computation. The maximum width of the interval is specified at computation time; however, due to the nature of the binary search algorithm, the computation may produce a narrower bound interval. All drop values for the results presented in this paper have been bounded to within an interval of width less than $10^{-6}$.

**Proposition 15.** *The calculated open bounding intervals for two drop values overlap if and only if the calculated bounds are identical.*

*Proof*: Suppose that the binary search for drop value $\mathfrak{D}_i(\lambda)$ has completed with bounds $l_i < \mathfrak{D}_i(\lambda) < u_i$, and that $i < n-1$. We consider the computation of $\mathfrak{D}_{i+1}(\lambda)$.

First, observe that no value of $l_i < p < u_i$ can have been tested in any of the previous binary search calculations. If they had been, then either the upper or lower bound of $\mathfrak{D}_i(\lambda)$ would have been updated at that time.

Second, observe that it must be the case that $l_i \leq l_{i+1}$. This is because $t_i \leq t_{i+1}$ by definition, and we know that $t_i^* = 1$ when $p = l_i$. So it must be the case that $t_{i+1}^* = 1$ when $p = l_i$ and so $l_i$ is a lower bound for $\mathfrak{D}_{i+1}(\lambda)$.

Now consider the value of $t_{i+1}^*$ when $p = u_i$. Note that this value would have been observed during the binary search for one of the previous drop values, and the starting bounds for $\mathfrak{D}_{i+1}(\lambda)$ would have been updated appropriately at that time.

- If $t_{i+1}^* = 1$ when $p = u_i$ then it must be the case that $u_i \le \mathfrak{D}_i(\lambda)$. Moreover, at the commencement of the binary search for $\mathfrak{D}_i(\lambda)$, the lower bound, $\mathcal{L}$ say, would satisfy $u_i \le \mathcal{L}$. So the binary search must produce bounds $l_{i+1}$ and $u_{i+1}$ such that $l_i < \mathfrak{D}_{i-1}(\lambda) < u_i \le l_{i+1} < \mathfrak{D}_i(\lambda) < u_{i+1}$ so the open intervals can not overlap.
- If $t_{i+1}^* < 1$ when $p = u_i$ then it must be the case that $\mathfrak{D}_{i+1}(\lambda) \le u_i$. Moreover, the first and second observations, above, imply that the starting bounds in the binary search must be $l_i = \mathcal{L} < \mathfrak{D}_{i+1}(\lambda) < \mathcal{U} = u_i$ The binary search for $\mathfrak{D}_i(\lambda)$ has completed, so the open interval $i < \mathfrak{D}_i(\lambda) < u_i$ is less than the required threshold. Thus the bounds $\mathcal{L} < \mathfrak{D}_{i+1}(\lambda) < \mathcal{U}$ are also less than the required threshold, and so the binary search will terminate immediately yielding $l_i = l_{i+1}$ and $u_i = u_{i+1}$.

We apply this reasoning iteratively starting at $i = 1$ and the result follows. □

Once we have bounded all $\mathfrak{D}_i(\lambda)$ we produce the optimization graph in parts. Each part is the interval between the upper bound $\mathfrak{D}_i(\lambda)$ to the lower bound of $\mathfrak{D}_{i+1}(\lambda)$ (inclusive). We do not need to compute $t_i^*$ for any value of $p$ less than $\mathfrak{D}_i(\lambda)$ because they are always 1.

For each part we use *Maple*'s plot function to plot $t_i^*$ for $p$ in the interval. It is a consequence of the plotting that the values of $t_i^*$ are calculated for values of $p$ within the interval, chosen by the plot function. We ensure these values of $p$ are chosen so that they are at most 0.005 apart, and that at least three are in the interval. Note that the $t_i^*$ are usually evaluated for significantly more values of $p$ than the minimum needed to fulfill this requirement, because the plot function employs an adaptive algorithm whereby it may choose additional values of $p$ so as to produce a smoother plot. When all the parts are plotted, we overlay them together on a single pair of axes to produce the graph.

**An important edge case.** Our implementation described in Implementation considerations does not work when $p = 1$. However, we observe that this case is precisely the case of a `PBP`. So values of $t_i^*$ for $p = 1$ are separately calculated using Eq (5) and are manually appended to the plot data to ensure the plot never attempts to evaluate $p = 1$ using our `C++` implementation.

Note that we optimize Eq (5) directly, instead of using $s_i^*$ from Eq (6) (the approximate optimal $t_i^*$ for Eq (5)) at the end of Optimal experimental design. Recall that the approximation is an asymptotic result in the number of observations. As such the utility of the approximation for the values of $n$ we use in this paper is poor.

The difference in Fisher information (Eq (8)) between using $s_i^*$ and using directly optimized $t_i^*$ is visualized in Fig 1. To constrain the size of the difference in the Fisher information we calculate the ratio of Fisher information from direct optimization to the Fisher information from the approximation, and subtract that ratio from one. Thus we achieved a clear visual indication of the percentage difference in Fisher information; a larger magnitude indicates a greater difference, a positive value indicates the optimized observation times produce a larger Fisher information (and thus are a more optimal set of observation times), and a negative value indicates that the Becker and Kersting approximations are more optimal than those produces by the numerical optimization.

We see that the directly optimized Fisher information is as much as 5% greater than the approximated optimal Fisher information when $n = 2$. Moreover, perhaps surprisingly, the difference in fact increases as $\lambda$ grows. Nonetheless, the maximum discrepancy does appear to shrink as $n$ grows, as we expect from the asymptotic result, decreasing to slightly under 1% when $n = 4$. We also see, unsurprisingly, that the approximated optimal observation times, $s_i^*$, are never more optimal than the numerically optimized observations times, $t_i^*$.

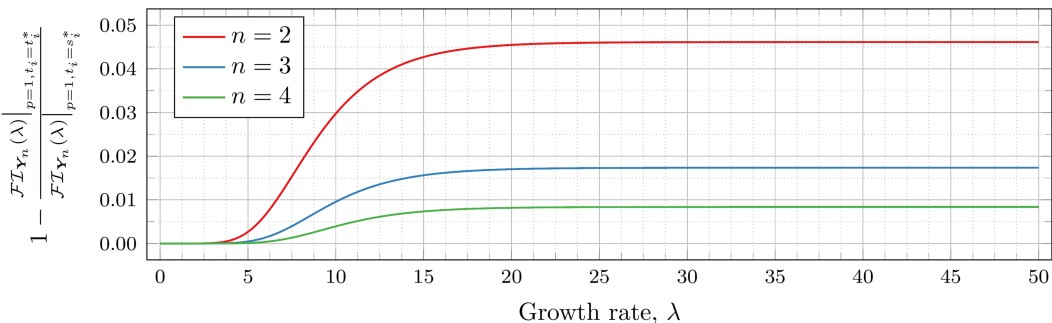

**Fig 1. Comparison of Fisher information for a PBP between directly optimized observation times and Becker & Kersting's approximate optimal observation times (Eq (6)).** Directly optimized: $t_i^*$, Becker & Kersting: $s_i^*$.

If we restrict our attention to only the ranges of $\lambda$ reported in our results ($0 \leq \lambda \leq 5$ for $n = 2$, $0 \leq \lambda \leq 4$ for $n = 3$ and $0 \leq \lambda \leq 1$ for $n = 4$), the discrepancy is much less pronounced. The results are shown in Fig 2, in which we also look at the difference between $s_i^*$ and $t_i^*$.

We see little noticeable difference in the Fisher information, but do see variance in the optimized observation times reach between 0.01 and 0.02 (which could affect the resulting parameter in the second or third decimal places). We note that although this difference is small, it was nonetheless large enough that if we used the approximation to compute $t_i^*$ for $p = 1$, we saw a noticeable "kink" in the graphs of optimal $t_i^*$ where the values that were numerically optimized from the POPBP Fisher information as $p \to 1$ met the value given by the approximation at $p = 1$.

Ultimately, optimizing Eq (5) directly is more accurate than using the Becker and Kersting approximation (i.e., Eq (6)). Consequently, we adopted that strategy for our computations.

## Experimental results

In this section we present the results of our computations. The presented results, along with the tools for producing them (predominantly *Maple* scripts, and some bash shell scripts) can be found in a Github repository (https://github.com/matt-sk/POPBP-Fisher-Information-Optimisation.git). Note that this repository is different from the repository for our C++ implementation, described above, of the Fisher information calculation. The results repository includes the C++ implementation repository as a submodule.

## Optimal observation times

Recall that $t_i^*$ are the optimal observation times that maximize the Fisher information, $p$ is the probability of an individual from the population being observed at any given time, $n$ is the number of observation times, and $\lambda$ is the growth rate of the population.

**Two observations ($n = 2$).** Figs 3, 4, 5, 6, 7, 8, and 9 show the values of $t_i^*$ as $p$ varies in the case of $n = 2$ observation times. They cover the cases of $\lambda = 0.5, 0.8, 1, 2, 3, 4,$ and $5$, respectively. Recall that $t_n^* = 1$ always, so the only $t_i^*$ in these figures shown is $t_1^*$.

When $\lambda = 0.5$ our calculations tell us that the drop value $\mathfrak{D}_1(0.5) \approx 0.98209$ (more precisely, the calculated bound was $0.982095718383789 < \mathfrak{D}_1(0.5) < 0.982096672058105$). Looking at Fig 3, the motivation for the name "drop value" should be apparent; we see that at $p \approx 0.98209$ the graph suddenly drops from $t_1^* = 1$. We indicate this drop with a dotted vertical line.

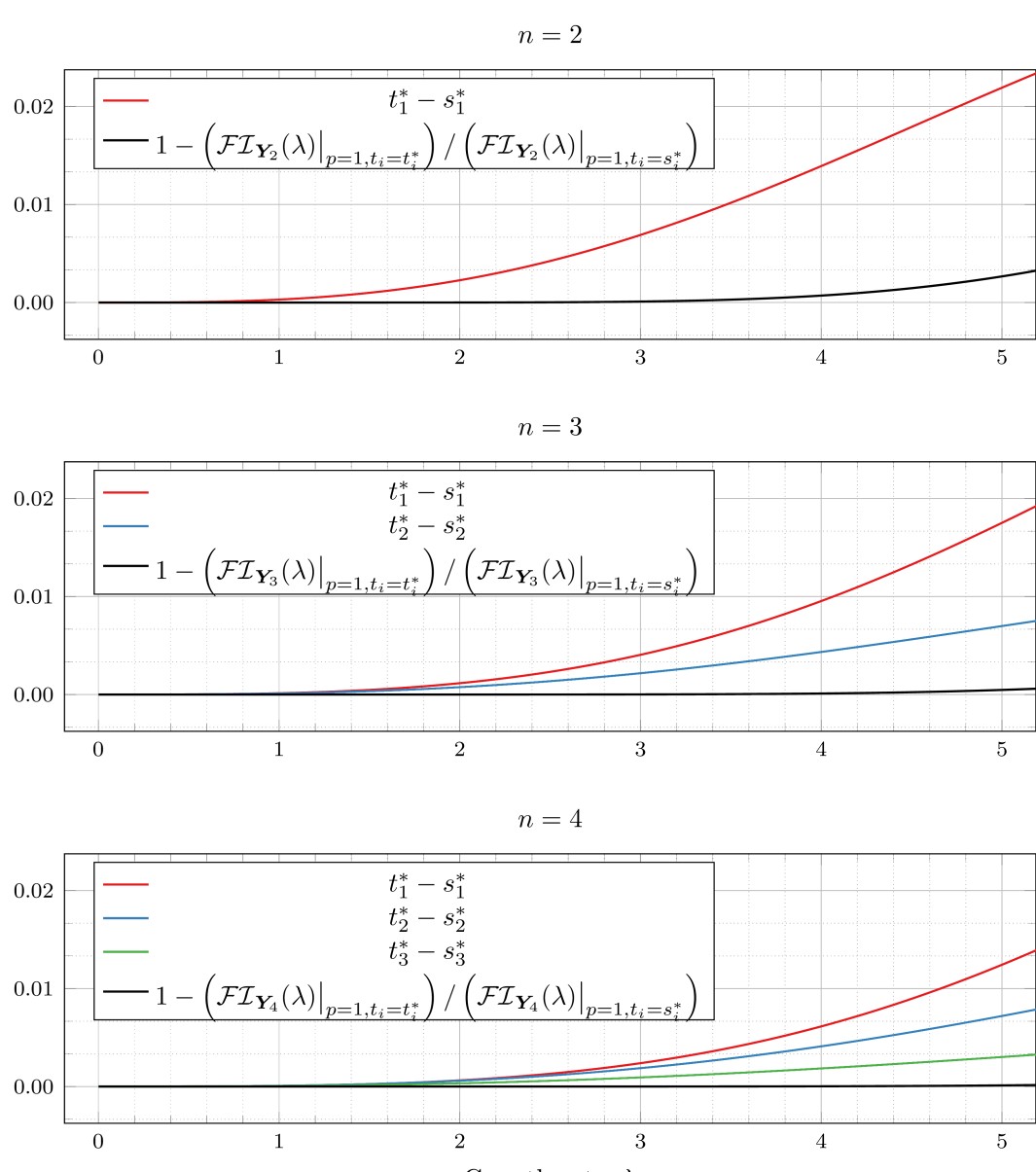

**Fig 2. Comparison of Fisher information and optimal observation times for a PBP between direct numerical optimization, and Becker & Kersting's approximate optimal observation times (Eq (6)).** Directly optimized: $t_i^*$, Becker & Kersting: $s_i^*$.

To understand why this drop happens, recall that lower values of $p$ mean the probability of individuals being observed at time $t_1$ is lower. To obtain a better estimator of $\lambda$, we want to be able to observe more individuals to more accurately gauge how much the population has grown (from $x_0 = 1$ individual initially). Waiting the maximum amount of time by taking the first (and second) observations at time 1 increases the expected number of individuals observed.

However, a point is reached where the information obtained from taking the first observation earlier outweighs the information obtained from taking the first observation at time

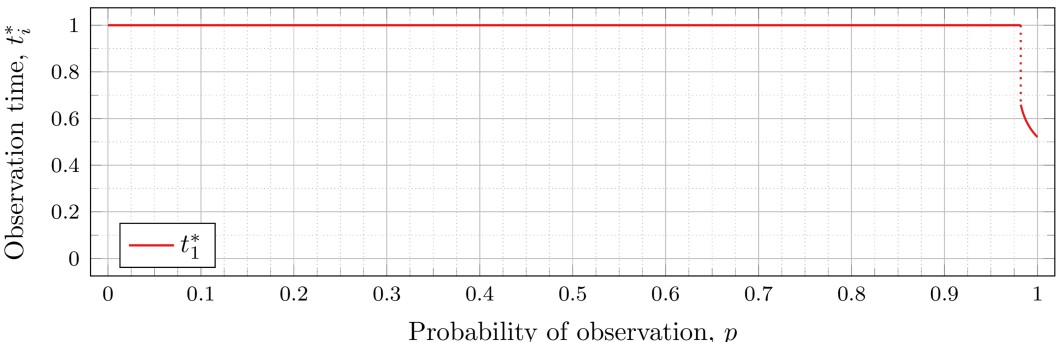

**Fig 3. Optimal observation times for $\mathcal{FI}_{Y_2}(0.5)$.**

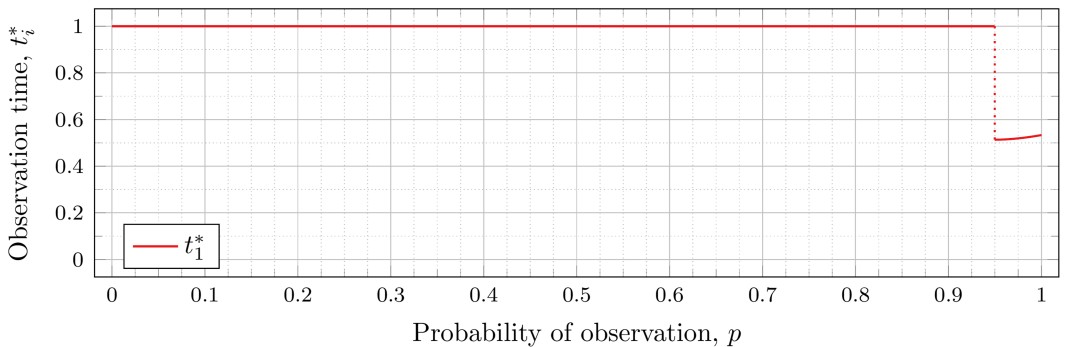

**Fig 4. Optimal observation times for $\mathcal{FI}_{Y_2}(0.8)$.**

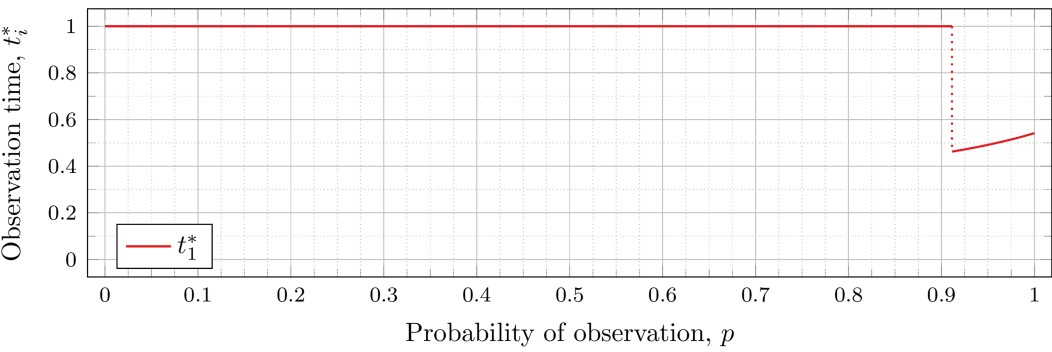

**Fig 5. Optimal observation times for $\mathcal{FI}_{Y_2}(1)$.**

$t_1 = 1$. This moment occurs precisely at the drop point (i.e., when $p = \mathfrak{D}_1(\lambda)$). Rather than decreasing smoothly from 1, $t_1^*$ drops instantly.

Note that in Fig 3 the values of $t_1^*$ decrease after the drop point. Conversely, for $\lambda = 0.8$ in Fig 4 the values increase after the drop point. This behavior continues for all subsequent values of $\lambda$ (for $n = 2$), although the curvature may vary.

We observe that the drop values decrease as $\lambda$ increases. Our computations tell us: $\mathfrak{D}_1(0.8) \approx 0.94976$, $\mathfrak{D}_1(1) \approx 0.91136$, $\mathfrak{D}_1(2) \approx 0.57116$, $\mathfrak{D}_1(3) \approx 0.26959$, $\mathfrak{D}_1(4) \approx 0.11468$,

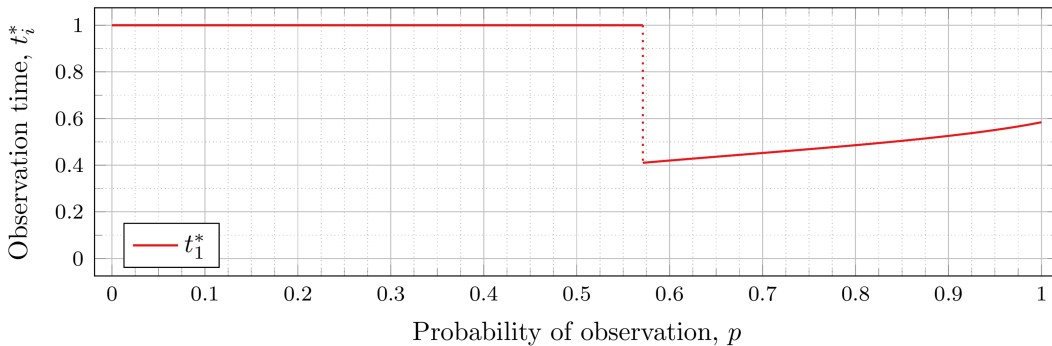

**Fig 6. Optimal observation times for $\mathcal{FI}_{Y_2}(2)$.**

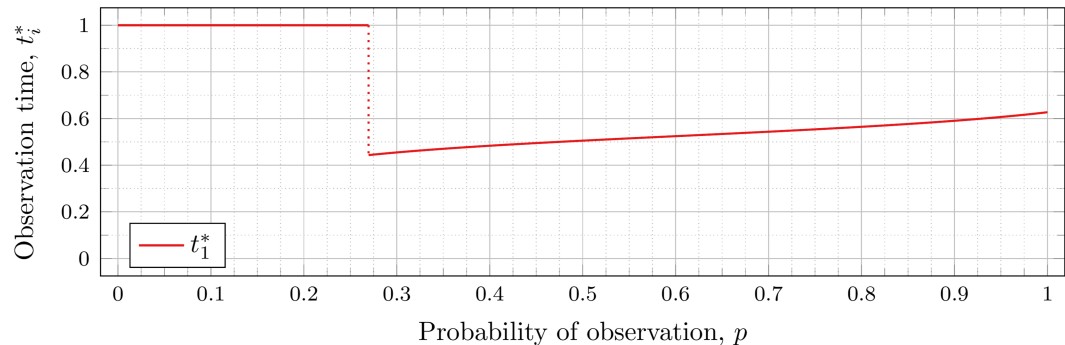

**Fig 7. Optimal observation times for $\mathcal{FI}_{Y_2}(3)$.**

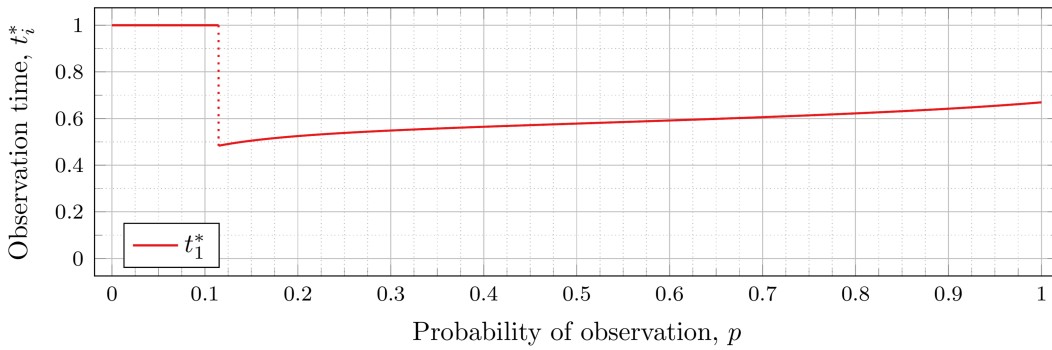

**Fig 8. Optimal observation times for $\mathcal{FI}_{Y_2}(4)$.**

and $\mathfrak{D}_1(5) \approx 0.048029$. We see drops in the graphs of Figs 4, 5, 6, 7, 8, and 9 at values of $p$ corresponding to these values. We see this pattern of decreasing drop values continues when $n = 3$ and $n = 4$.

To visualize the behavior of the change in $\mathfrak{D}_1(\lambda)$, we plot them against $\lambda$ in Fig 10. We see that the decrease is not linear.

Recall that our computations bound the drop value. To speed up the computation time for the graph in Fig 10, we computed for each $\lambda$ until the upper and lower bounds for $\mathfrak{D}_1(\lambda)$

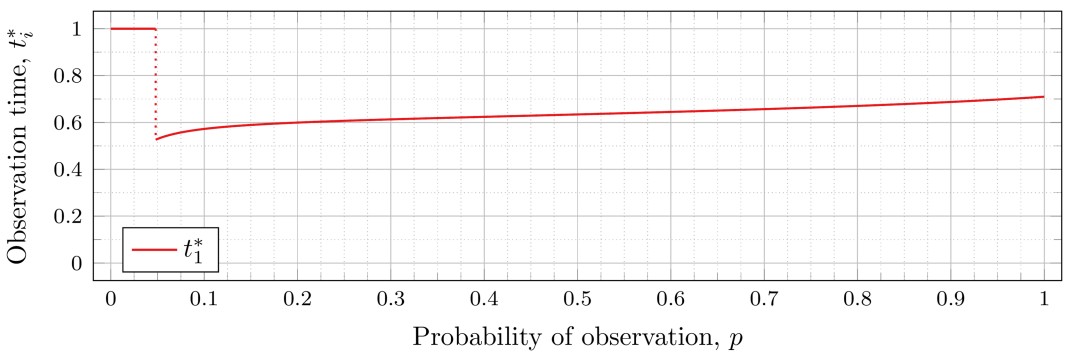

**Fig 9. Optimal observation times for $\mathcal{FI}_{Y_2}(5)$.**

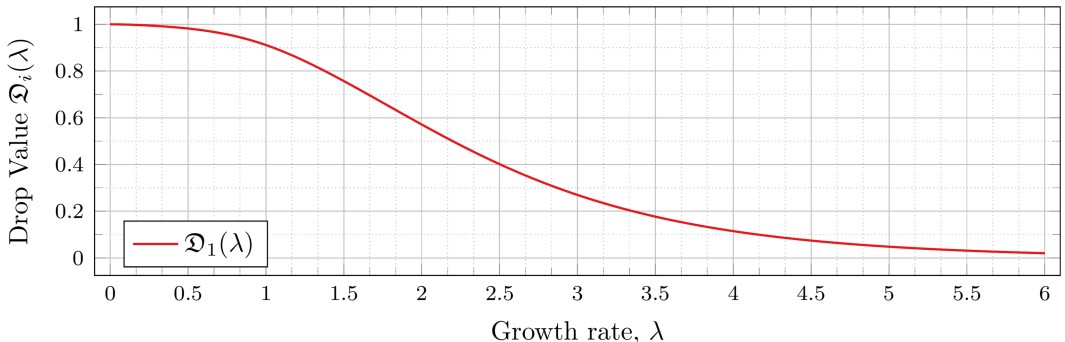

**Fig 10. Change in drop values for $n = 2$ as $\lambda$ varies.**

were less than $10^{-3}$ apart (instead of $10^{-6}$ as used in our other drop-value calculations). Two hundred values of $\lambda$ were used to generate the graph. Note that we have plotted both the upper and lower bounds in Fig 10; however, the bounds are sufficiently close that the difference is unable to be discerned in the graph.

This behavior can be explained by recalling that higher $\lambda$ means a higher population growth rate, so the population grows relatively larger (compared with a population with a lower growth rate) over time. This growth results in a lower probability $p$ required to obtain a satisfactory expected number of observed individuals at time $t_1^*$ (i.e., a lower drop value).

We see, in Fig 11, the Fisher information plotted against $t_1$ and $p$, for the case of $n = 2$ observations and the growth rates of $\lambda = 1, 2, 3$. We have additionally overlaid the optimal Fisher information onto the graph (in a thick black line on each surface). If we observe each plot looking down (observing only the $t_1^*$–$p$ plane) we see in the black line (the optimal Fisher information) precisely the shape of the curves in Figs 5, 6, and 7, respectively. If we observe each plot looking only at the $FI^*$–$p$ plane we see in the black line (the optimal Fisher information) precisely the shape of the optimal Fisher information as seen in the corresponding optimal Fisher information graphs we present in Optimal Fisher information later in this section.

Recall that Bean et al. [15] could not calculate the Fisher information for high values of $\lambda$ (even for the $n = 2$ case), and thus could not assess the quality of the $\widetilde{\mathcal{FI}}_{Y_2}(\lambda)$ approximation for such $\lambda$. However, as our new approach allows us to calculate $\mathcal{FI}_{Y_2}(\lambda)$ for high values of

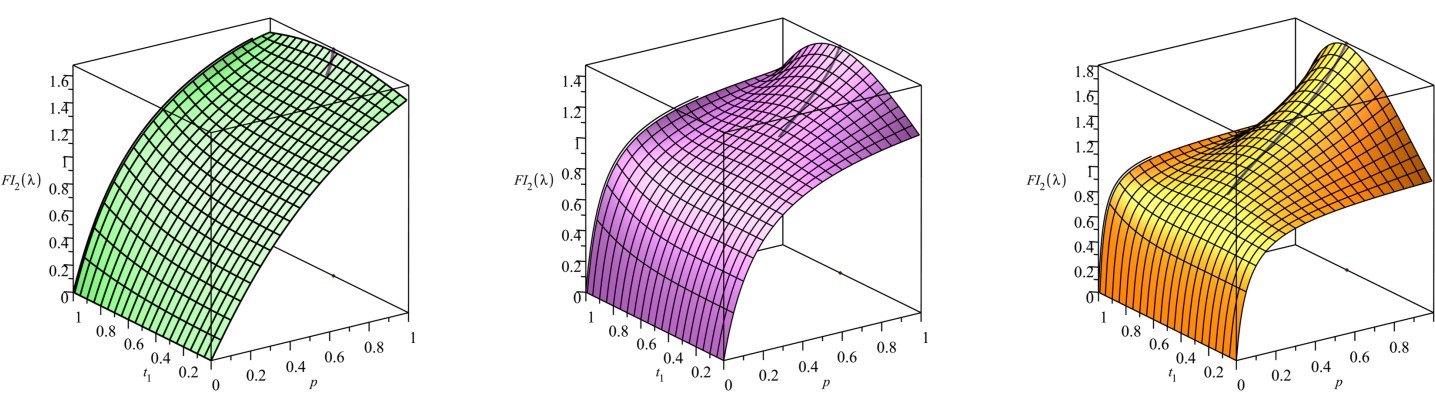

**Fig 11. Optimal Fisher information for $n = 2$.** Plotted against $t_1$ and $p$. Left: $\lambda = 1$, Mid: $\lambda = 2$, Right: $\lambda = 3$.

$\lambda$ more efficiently, we can use it to assess the quality of $\widetilde{\mathcal{FI}}_{\mathbf{Y}_2}(\lambda)$ for finding optimal observation times for higher values of $\lambda$. To this end, let $\widetilde{t_i^*}$ denote the optimal observation times for $\widetilde{\mathcal{FI}}_{\mathbf{Y}_2}(\lambda)$. We show in each of Figs 12, 13, 14, 15, 16, 17, and 18 both $\widetilde{t_i^*}$ and $t_i^*$ (each for a different value of $\lambda$).

We computed the values of $\widetilde{t_1^*}$ using the same optimization libraries in *Maple* described above, but using a symbolic representation of $\widetilde{\mathcal{FI}}_{\mathbf{Y}_2}(\lambda)$ instead of our C++ Fisher information

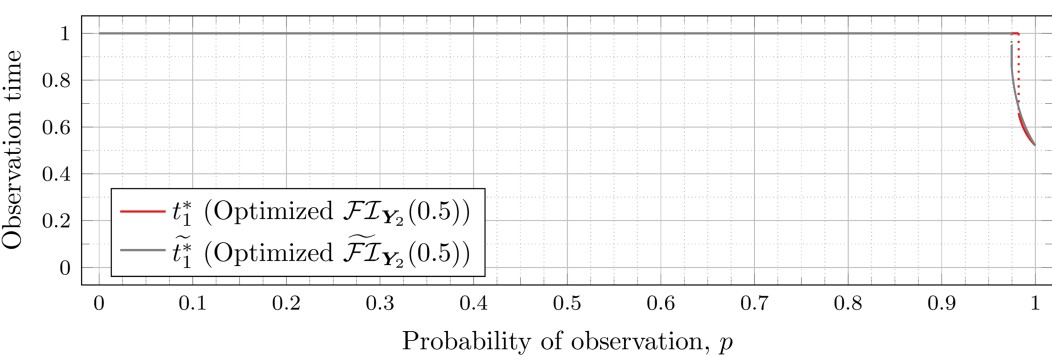

**Fig 12. Optimal observation times for $\widetilde{\mathcal{FI}}_{\mathbf{Y}_2}(0.5)$ and $\mathcal{FI}_{\mathbf{Y}_2}(0.5)$.**

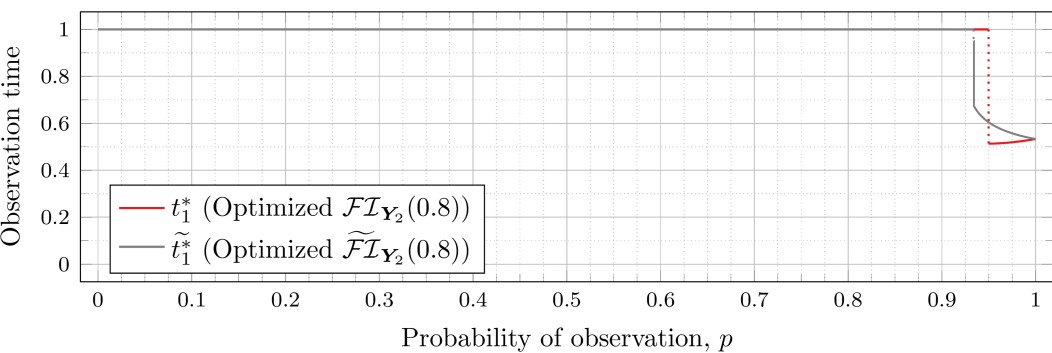

**Fig 13. Optimal observation times for $\widetilde{\mathcal{FI}}_{\mathbf{Y}_2}(0.8)$ and $\mathcal{FI}_{\mathbf{Y}_2}(0.8)$.**

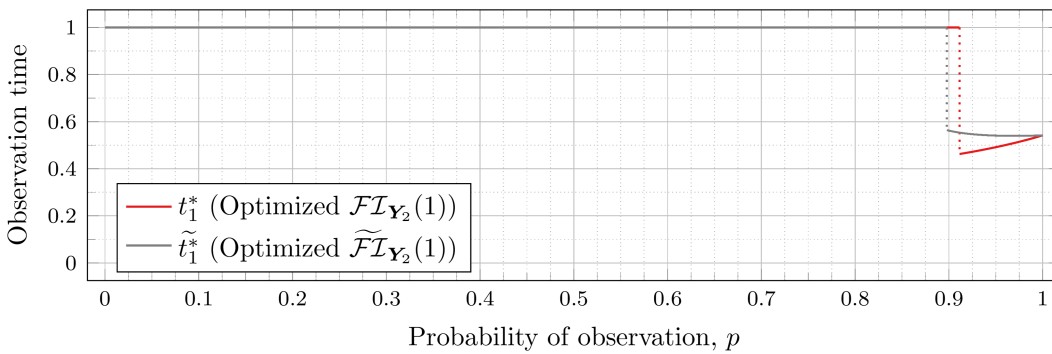

**Fig 14. Optimal observation times for $\widetilde{\mathcal{FI}}_{Y_2}(1)$ and $\mathcal{FI}_{Y_2}(1)$.**

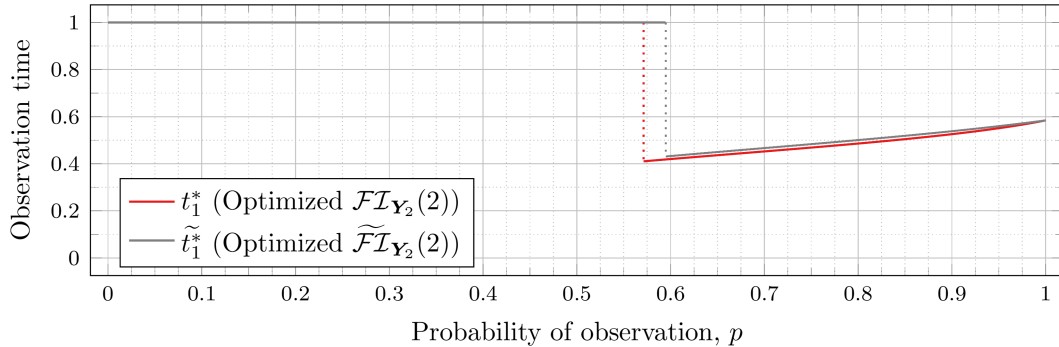

**Fig 15. Optimal observation times for $\widetilde{\mathcal{FI}}_{Y_2}(2)$ and $\mathcal{FI}_{Y_2}(2)$.**

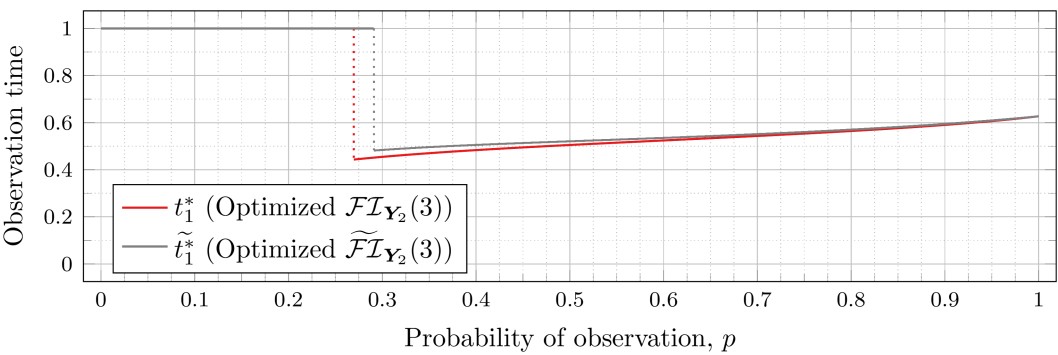

**Fig 16. Optimal observation times for $x\widetilde{\mathcal{FI}}_{Y_2}(3)$ and $\mathcal{FI}_{Y_2}(3)$.**

calculation implementation. Undefined values exist in Eq (10) when $t_1 = t_2$, $t_1 = 0$, $t_2 = 0$, and $t_1 = t_2 = 0$. We accounted for these by pre-computing in *Maple* the limits of the formula for all of these cases (i.e., as $t_2 \to t_1$, $t_1 \to 0$, $t_2 \to 0$, and $(t_1, t_2) \to (0, 0)$) and employed a piecewise function to choose the correct expression.

The aforementioned piecewise function evaluates the equality of $t_1$ and $t_2$ by checking if $t_1 - t_2 = 0.0$ (i.e., numeric zero instead of symbolic zero) to account for cases where the values

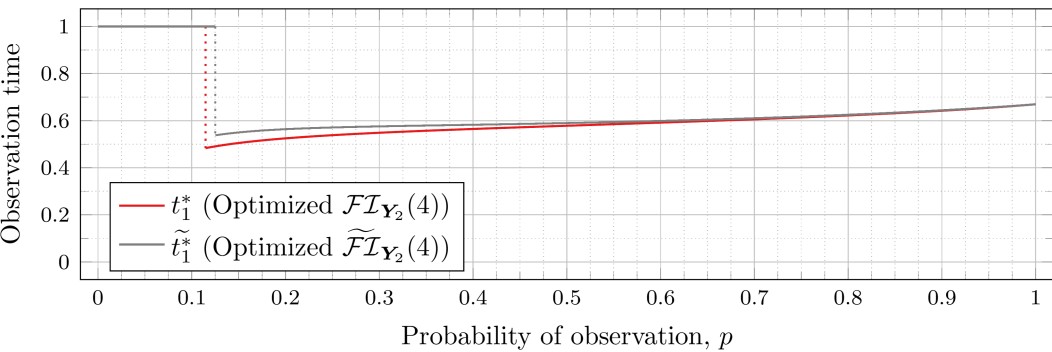

**Fig 17. Optimal observation times for $\widetilde{\mathcal{FI}}_{Y_2}(4)$ and $\mathcal{FI}_{Y_2}(4)$.**

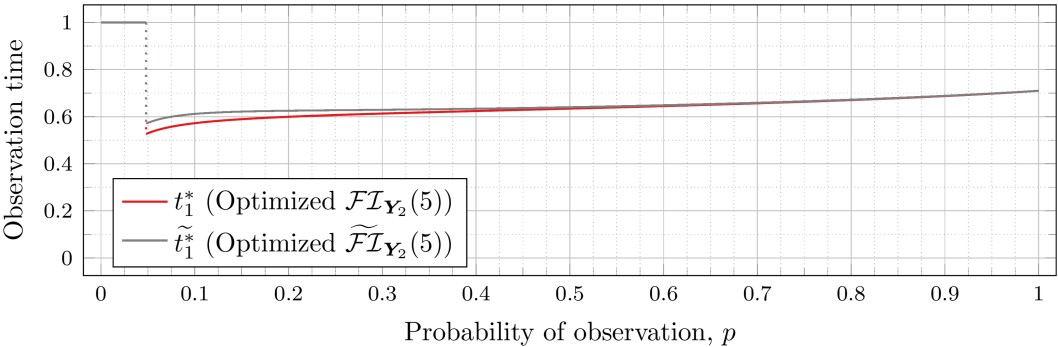

**Fig 18. Optimal observation times for $\widetilde{\mathcal{FI}}_{Y_2}(5)$ and $\mathcal{FI}_{Y_2}(5)$.**

are not precisely the same, but are sufficiently close to yield the undefined values. We similarly check for $t_1$ and $t_2$ being equal to numeric zero.

Note that the graphs in Figs 12 and 13 have a vertical gray line immediately after the dotted gray line indicating the sudden $t_1^*$ drop. These vertical lines are due to the optimization routines finding values of $t_1^* = 0.95$ for some $p$ values immediately after $\mathfrak{D}_1(\lambda)$ but before a second (more significant) drop. The effect is more apparent visually in Fig 13. The remaining graphs do not exhibit this behavior.

In the case of $\lambda = 0.5$ for $\widetilde{\mathcal{FI}}_{Y_2}(0.5)$, we have $\mathfrak{D}_1(0.5) \approx 0.97448$. After the drop point, $t_1^*$ drops to 0.95 until $p \approx 0.97466$ at which point it drops again to $t_1^* \approx 0.85854$. Comparatively, for $\mathcal{FI}_{Y_2}(0.5)$, we have $\mathfrak{D}_1(0.5) \approx 0.98209$ after which $t_1^* \approx 0.65952$. Thus the graph of $\widetilde{\mathcal{FI}}_{Y_2}(0.5)$ doesn't drop as far as $\mathcal{FI}_{Y_2}(0.5)$; nonetheless, the curves are remarkably close.

In the case of $\lambda = 0.8$ for $\widetilde{\mathcal{FI}}_{Y_2}(0.8)$, we have $\mathfrak{D}_1(0.8) \approx 0.93391$. After the drop point, $t_1^*$ drops to 0.95 until $p \approx 0.93422$ at which point it drops again to $t_1^* \approx 0.67365$. Comparatively, for $\mathcal{FI}_{Y_2}(0.8)$, we have $\mathfrak{D}_1(0.8) \approx 0.94976$ after which $t_1^* \approx 0.51352$.

In all cases, we see both curves approach the same $t_1^*$ when $p = 1$. This observation is clearest when $\lambda = 0.8$ and $\lambda = 1$, for which the approximation $\widetilde{\mathcal{FI}}_{Y_2}(\lambda)$ appears to be poorest. The reason is that when $p = 1$, the POPBP$(1, \lambda)$ reduces to the PBP$(\lambda)$, and both $\widetilde{\mathcal{FI}}_{Y_2}(\lambda)$ and $\mathcal{FI}(\lambda)$ coincide with the exact form of the Fisher information for a PBP as given in Eq (5). We note that we did not force the computation of Eq (5) for $p = 1$ when computing $\widetilde{\mathcal{FI}}_{Y_2}(\lambda)$ like we did for the computation of $\mathcal{FI}_{Y_2}(\lambda)$, and yet the curves nonetheless meet.

In all cases, we see that as $p$ increases, the two curves ($\widetilde{\mathcal{FI}}_{Y_2}(\lambda)$ and $\mathcal{FI}(\lambda)$) become closer together. Fig 12 notwithstanding, as $\lambda$ increases the curves become closer together as well. Therefore, when $n = 2$, we can use the approximation $\widetilde{\mathcal{FI}}_{Y_2}(\lambda)$ to find $t_1^*$ when $\lambda$ is large, because the calculation is faster and the approximation is quite good. When $\lambda$ is small, we can find $t_1^*$ directly, because the calculation time is shorter, and $\widetilde{\mathcal{FI}}_{Y_2}(\lambda)$ is a poorer approximation.

**Three observations ($n = 3$).** Figs 19, 20, 21, 22, 23, and 24 show the values of $t_i^*$ as $p$ varies in the case of $n = 3$ observation times. They cover the cases of $\lambda = 0.5, 0.8, 1, 2, 3$, and $4$, respectively. Recall that $t_n^* = 1$ always, so only $t_1^*$ and $t_2^*$ are shown in these figures.

In the $\lambda = 3$ case, the end of the curve for $t_1^*$ before $\mathfrak{D}_2(3)$ appears to meet the beginning of the curve of $t_2$ after $\mathfrak{D}_2(3)$. Looking more closely at this region (as shown in Fig 25) shows that although they are very close, the curves do not meet.

Observe that in all the figures, the graphs of $t_1^*$ drop a second time at $p \approx \mathfrak{D}_2(\lambda)$. We did not explicitly calculate these second (nor third, etc) drops for $t_1^*$, yet they nonetheless appear in the calculated graphs. Recall that $\mathfrak{D}_i(\lambda)$ is specifically defined as (and thus computed to be) the value for the *first* time $t_i^*$ drops. Consequently, $\mathfrak{D}_2(\lambda)$ is first time that $t_2^*$ drops, and we stress that none of the computations for its bounds took $t_1^*$ into account at all.

Furthermore, we do not compute any values of $t_i^*$ (for any $i$) for values of $p$ within the bounds of $\mathfrak{D}_j(\lambda)$ (for any $j$). Thus whether $t_1^*$ drops at exactly $p = \mathfrak{D}_2(\lambda)$, or at a point very close, is unclear. However, we note that an earlier calculation had $\mathfrak{D}_i(\lambda)$ bounded by an interval of width less than $10^{-12}$, and we observed the same phenomenon in the graphs calculated

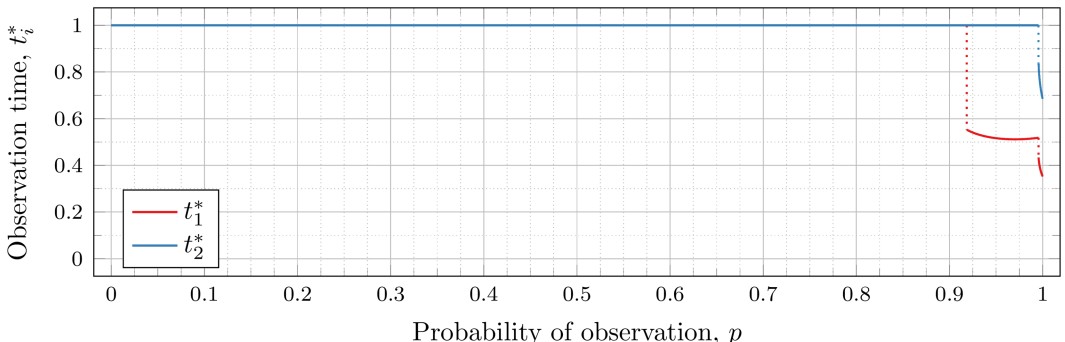

**Fig 19. Optimal observation times for $\mathcal{FI}_{Y_3}(0.5)$.**

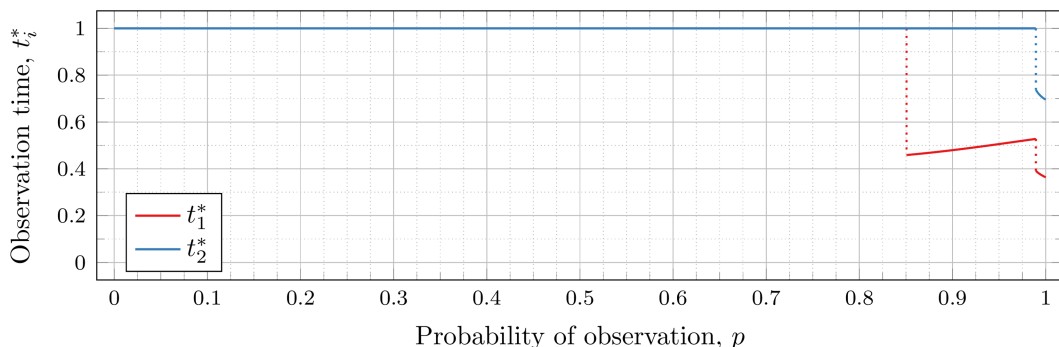

**Fig 20. Optimal observation times for $\mathcal{FI}_{Y_3}(0.8)$.**

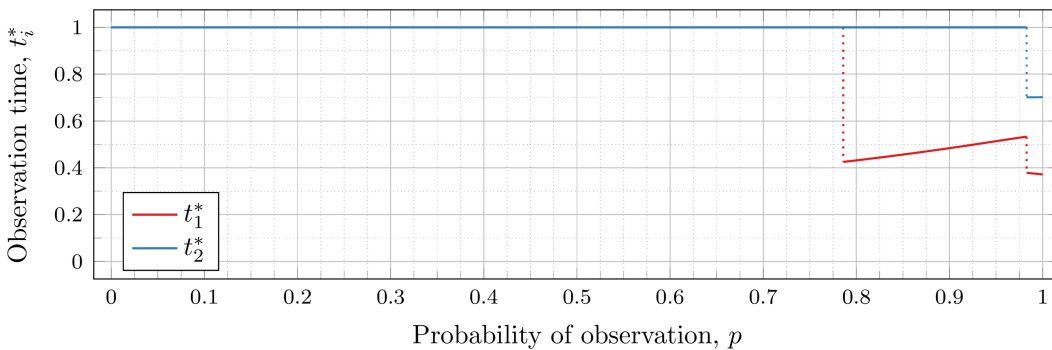

**Fig 21. Optimal observation times for $\mathcal{FI}_{Y_3}(1)$.**

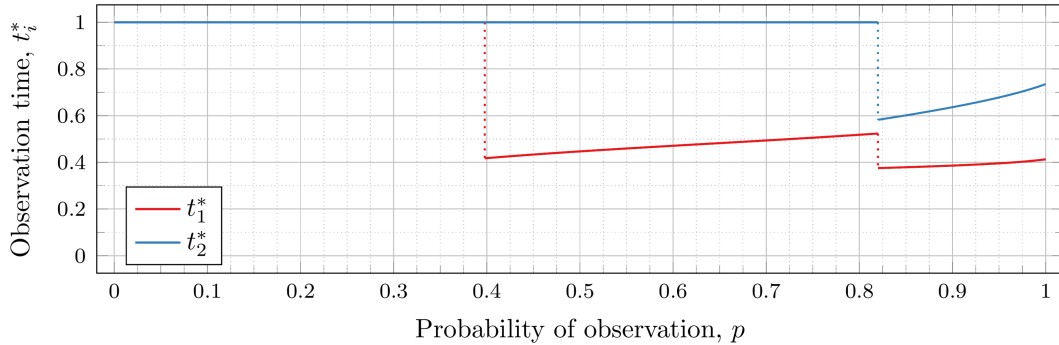

**Fig 22. Optimal observation times for $\mathcal{FI}_{Y_3}(2)$.**

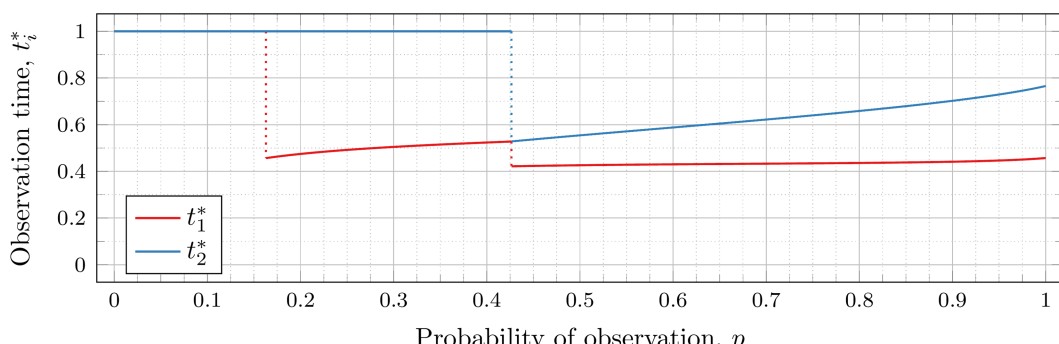

**Fig 23. Optimal observation times for $\mathcal{FI}_{Y_3}(3)$.**

at that time. We abandoned and recomputed the data because we had not taken timing information for the calculations, and, importantly, the timing information reported in this paper is for the computations used to produce the results reported herein. When recomputing, we decided that bounding $\mathfrak{D}_i(\lambda)$ to an interval less than $10^{-12}$ was overkill, and we opted to use $10^{-6}$ instead.

We see this behavior consistently in all our graphs for $n = 3$ and $n = 4$. In general, $t_i^*$ for all $i \leq j$ suddenly drop in value at $p \approx \mathfrak{D}_j(\lambda)$.

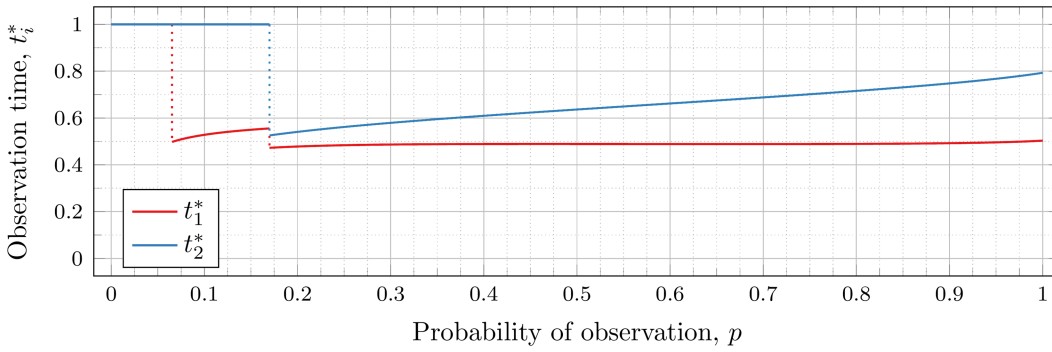

**Fig 24. Optimal observation times for $\mathcal{FI}_{Y_3}(4)$.**

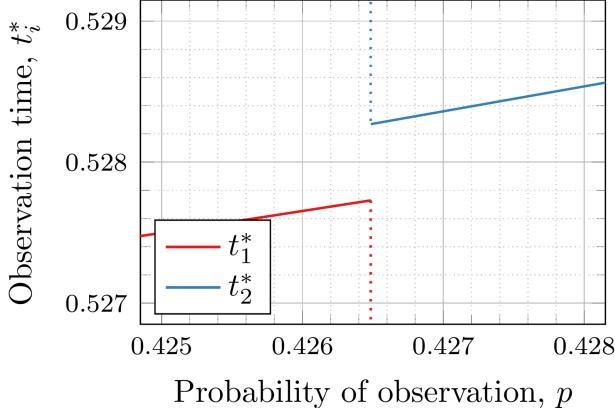

**Fig 25. Optimal observation times for $\mathcal{FI}_{Y_3}(3)$.** Plot zoomed in to show the curves do not touch.

In all cases, we see the same phenomena with regard to drop values that we do for the $n = 2$ case. As $\lambda$ increases, both drop values ($\mathfrak{D}_1(\lambda)$ and $\mathfrak{D}_2(\lambda)$) decrease. Unsurprisingly, $\mathfrak{D}_1(\lambda)$ is always less than $\mathfrak{D}_2(\lambda)$; however, the distance between them varies as $\lambda$ increases.

To visualize the behavior of the change in both $\mathfrak{D}_1(\lambda)$ and $\mathfrak{D}_2(\lambda)$ we plot them against $\lambda$ in Fig 26. As we did for the $n = 2$ case, we computed until the bounding intervals were less than $10^{-3}$, and plot both upper and lower bounds in the figure for each $\mathfrak{D}_i(\lambda)$.

**Four observations ($n = 4$).** Figs 27, 28, and 29 show the values of $t_i^*$ as $p$ varies in the case of $n = 4$ observation times. They cover the cases of $\lambda = 0.5, 0.8$ and 1, respectively. Recall that $t_n^* = 1$ always, so only $t_1^*$, $t_2^*$, and $t_3^*$ are shown in these figures.

In all cases, we see the same phenomena with regard to drop values that we did for the $n = 2$ and $n = 3$ case. As $\lambda$ increases, all drop values ($\mathfrak{D}_1(\lambda)$, $\mathfrak{D}_2(\lambda)$, and $\mathfrak{D}_3(\lambda)$) decrease. Unsurprisingly, $\mathfrak{D}_1(\lambda)$ is always less than $\mathfrak{D}_2(\lambda)$ which in turn is always less than $\mathfrak{D}_3(\lambda)$; however, the distance between them varies as $\lambda$ increases.

We did not plot $\mathfrak{D}_i(\lambda)$ against $\lambda$ to visualize the behavior of the change in the drop values. We chose not to do so because of the large time required to compute the drop values for a single $\lambda$ (even when only computing to an interval less than $10^{-3}$), and the large number of values of $\lambda$ required to produce such a plot.

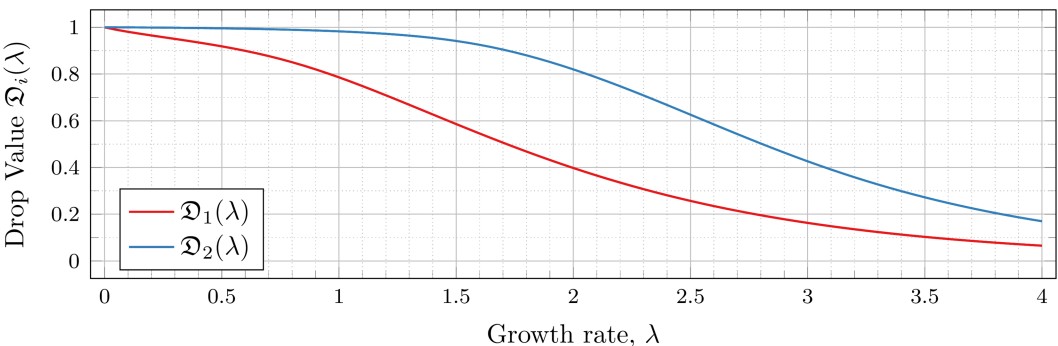

**Fig 26. Change in drop values for $n = 3$ as $\lambda$ varies.**

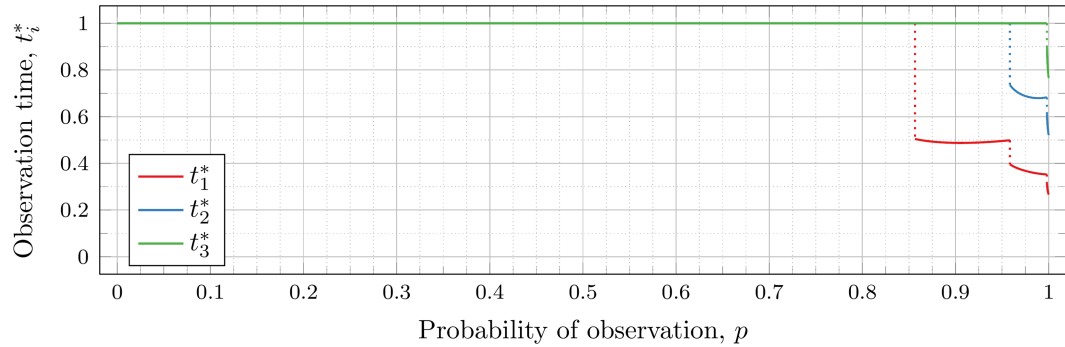

**Fig 27. Optimal observation times for $\mathcal{FI}_{Y_4}(0.5)$.**

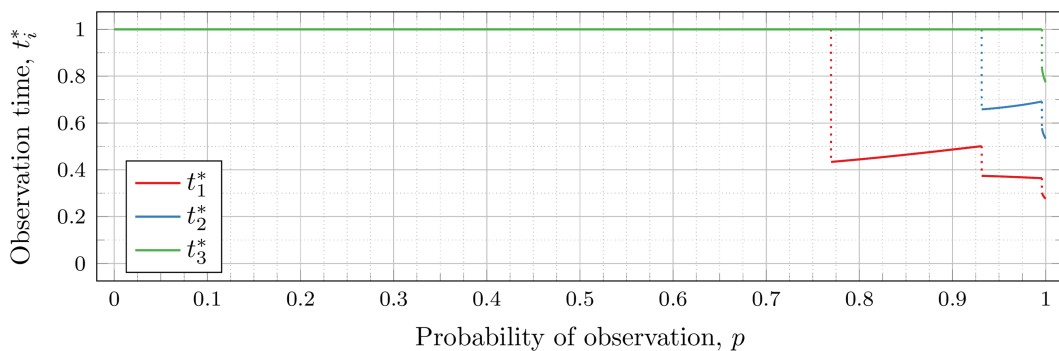

**Fig 28. Optimal observation times for $\mathcal{FI}_{Y_4}(0.8)$.**

### Optimal Fisher information

Figs 30, 31, 32, 33, 34, 35, and 36 show the optimal Fisher information, $\mathcal{FI}^*$, corresponding to the optimal parameters $t_i^*$ from the 'Two observations ($n = 2$)', 'Three observations ($n = 3$)', and 'Four observations ($n = 4$)' subsections, above.

In all cases the graph appears to be increasing. We see an initial concave-down curve followed by a sudden change of curvature and direction in the graph (and an undifferentiable point at the interface between the two). Careful observation should indicate that this change

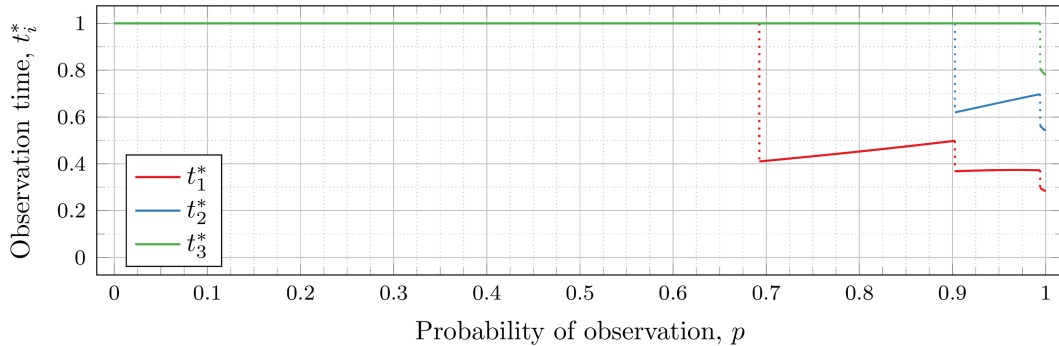

**Fig 29. Optimal observation times for $\mathcal{FI}_{Y_4}(1)$.**

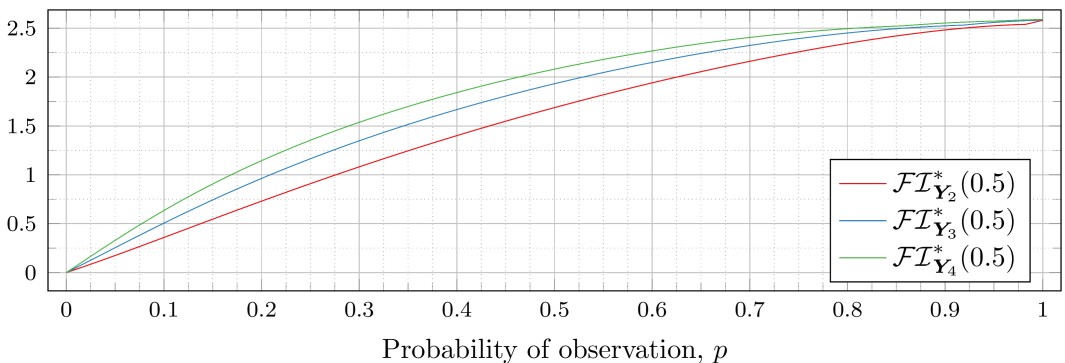

**Fig 30. Optimal Fisher information for $\lambda = 0.5$.**

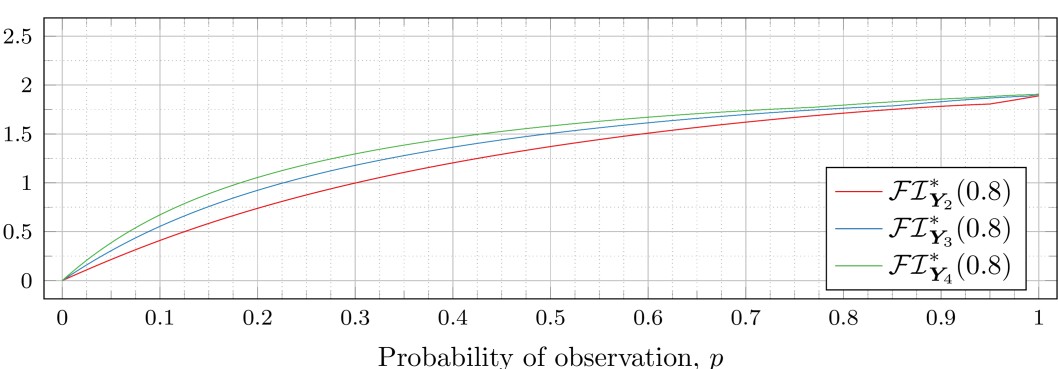

**Fig 31. Optimal Fisher information for $\lambda = 0.8$.**

corresponds to a drop value $\mathfrak{D}_i(\lambda)$. Indeed, the curves for $n = 2$ have only a single change, whereas the curves for $n = 3$ have two changes (although in some cases seeing the second change can be difficult), and for $n = 4$ we have three changes.

As should be expected, for a fixed growth rate $\lambda$ we see the optimal Fisher information increases as the number of observations, $n$, increase. For small $\lambda$ the curves appear to converge to the same value as $p$ approaches 1; however, convergence ceases when $\lambda \geq 2$.

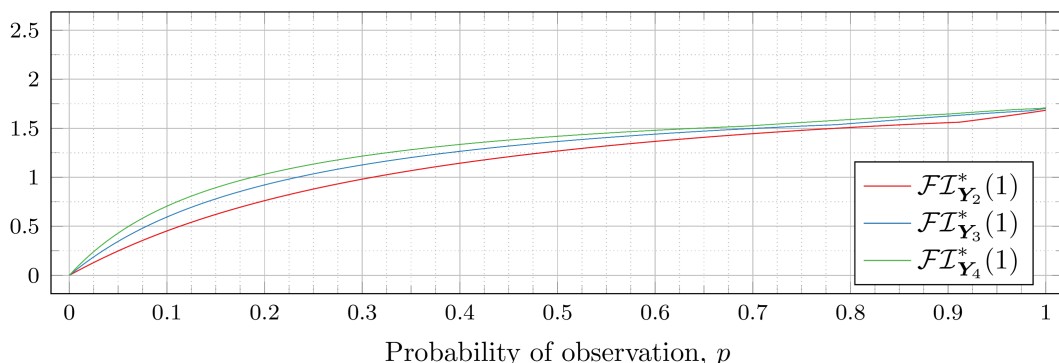

**Fig 32. Optimal Fisher information for** $\lambda = 1$**.**

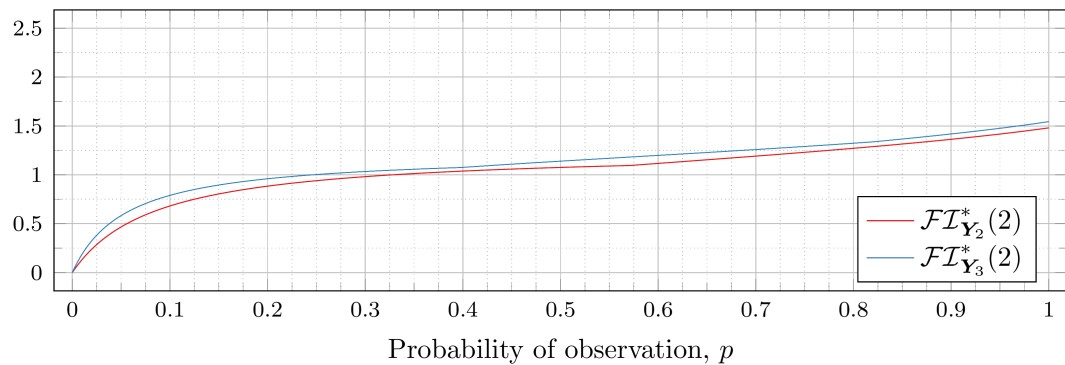

**Fig 33. Optimal Fisher information for** $\lambda = 2$**.**

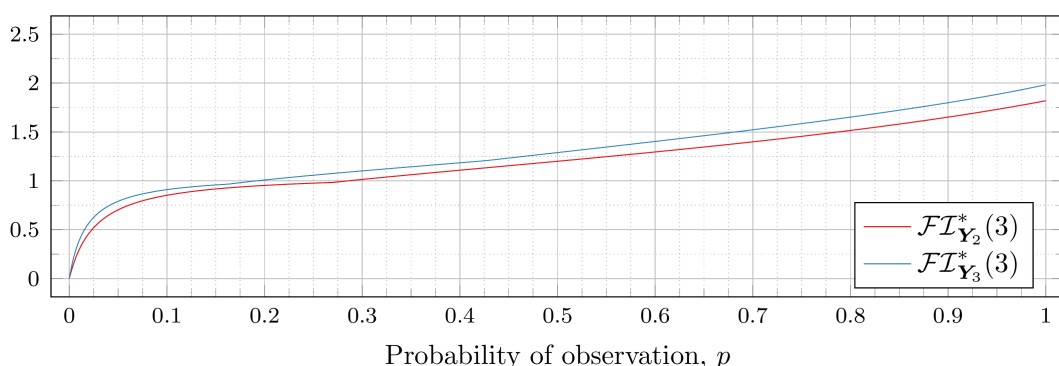

**Fig 34. Optimal Fisher information for** $\lambda = 3$**.**

Furthermore, for a fixed growth rate $\lambda$ and number of observations $n$, the optimal Fisher information increases as the probability of observation $p$ increases.

A surprising observation is that the maximal optimal Fisher information decreases as $\lambda$ increases from 0.5 to 2, which can be seen by comparing the optimal Fisher information for $p = 1$ on each graph and observing that it is decreasing. This observation is clearer when we draw the graphs side by side on equally sized axes as shown in Figs 37 and 38.

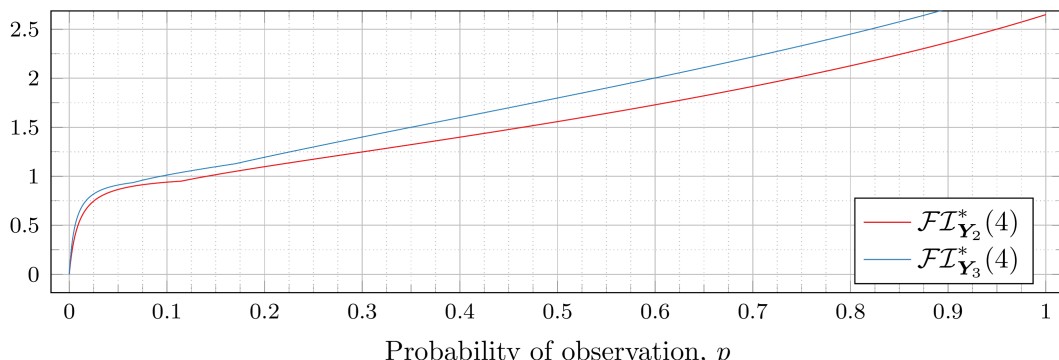

**Fig 35. Optimal Fisher information for $\lambda = 4$.**

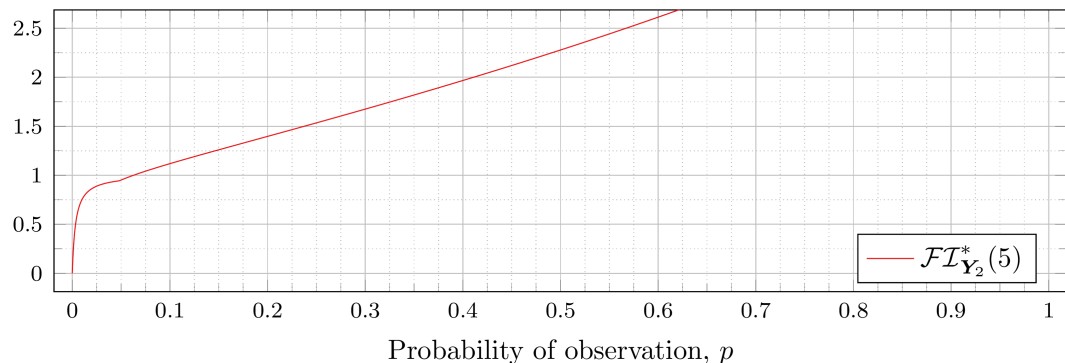

**Fig 36. Optimal Fisher information for $\lambda = 5$.**

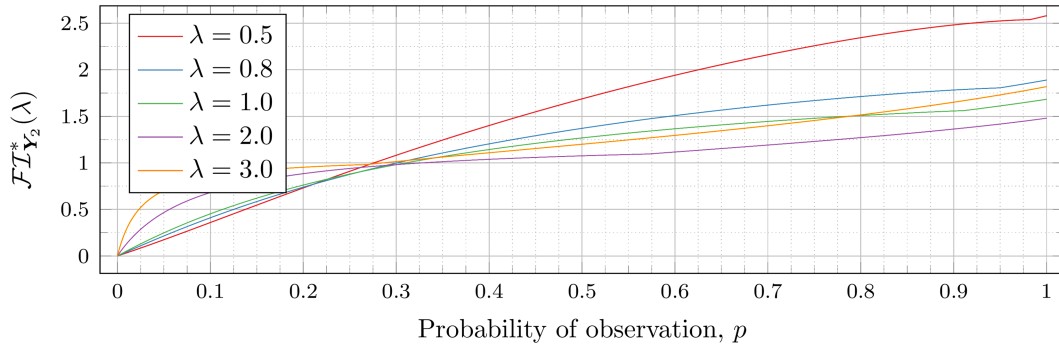

**Fig 37. Comparison of Optimal Fisher information for $n = 2$.**

## Timings

All computations reported in this section were performed on Intel(R) Xeon(R) Gold 6150 CPU's running at 2.70 GHz. All computations for $n = 2$ were performed single-threaded, whereas all computations for $n \geq 3$ were performed multi-threaded using 36 simultaneous threads.

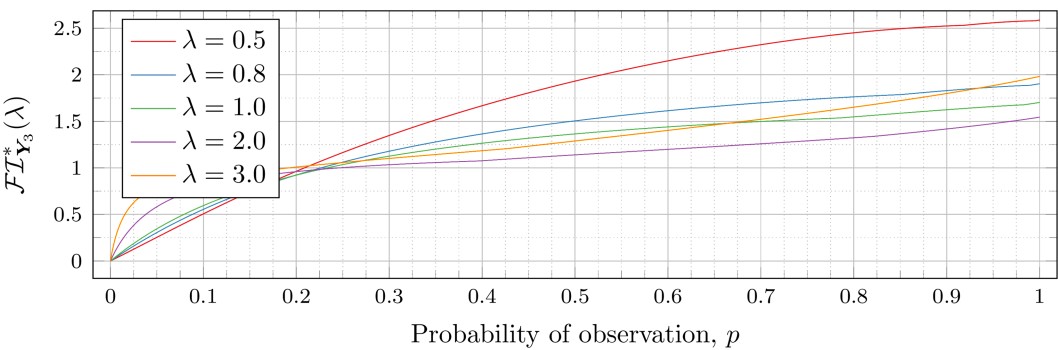

**Fig 38. Comparison of Optimal Fisher information for *n* = 3**.

The reason for the single-threaded computations for *n* = 2 is that we found the multi-threaded implementation was slower than the single-threaded implementation for small values of $\lambda$ (when *n* = 2).

An individual computation of $\mathcal{FI}_{Y_n}(\lambda)$ was so fast that the time required to initialize the threading requirements was significant enough to overshadow the time required to compute with a single thread. As such, and because running all computations for a single *n* with the same *Maple* scripts was easier, all computations for *n* = 2 were performed single-threaded even though some of these computations might have benefited from multi-threading.

We note that the scheduling software placed an upper limit of 300 hours (12.5 days) on a single computation. The only computation for which this 300 hour limitation was a concern was the computation of the graph data for *n* = 3 and $\lambda$ = 4.0. The code was written to allow a terminated computation to recommence from the point of termination; however, the saved data that allowed for recommencement was at the granularity of completed optimizations. Consequently, the optimization being performed when the computation terminated at the 300-hour boundary needed to begin again when the computation resumed in the next 300 hour block, and so some amount of re-computation was unavoidable.

The optimization time for $\mathcal{FI}_{Y_n}(\lambda)$ grows as *n* increases. Furthermore, for fixed *n*, the optimization times increase as $\lambda$ increases and as *p* increases.

The times taken to bound $\mathfrak{D}_i(\lambda)$ in an open interval of width less than $10^{-6}$ are given in Table 2. We show the total time, the number of optimizations performed (i.e., different values of *p*) and the fastest, slowest, and average times for individual optimizations.

Note that because of the nature of the binary search, the optimizations will tend to clump around the drop values. If those drop values are small, we will be performing more of the faster optimizations. Conversely if the drop values are large, we will be performing more of the slower optimizations. As such, the table should not be taken as a strong indication of the relative speed of the computation times for different parameters *n* and $\lambda$.

Also note that because we have *n*−1 drop values, the number of optimizations performed must grow as *n* increases.

The time taken to compute the data required to produce the graphs of $t_i^*$ in Experimental results are given in Table 3. Note that more optimizations are performed (compared with the drop-value computation); however, we only optimized for values of *p* between the drop values. In particular, no values of *p* before $\mathfrak{D}_1(\lambda)$ are ever calculated.

Furthermore, as $\lambda$ increases, we see (as discussed above) a decrease in the drop values, and an increase in the size of the intervals between the drop values. Consequently, more

**Table 2. Computation times for $\mathfrak{D}_i(\lambda)$.**

| n | $\lambda$ | Time | #Opt | Fastest | Slowest | Average |
|---|---|---|---|---|---|---|
| 2 | 0.5 | 0.8s | 20 | 0.0s | 0.0s | 0.0s |
| 2 | 0.8 | 1.5s | 20 | 0.0s | 0.1s | 0.1s |
| 2 | 1.0 | 2.4s | 20 | 0.0s | 0.2s | 0.1s |
| 2 | 2.0 | 8.2s | 20 | 0.3s | 0.8s | 0.4s |
| 2 | 3.0 | 15.4s | 20 | 0.6s | 1.9s | 0.8s |
| 2 | 4.0 | 29.0s | 20 | 0.3s | 11.3s | 1.4s |
| 2 | 5.0 | 2m 29.7s | 20 | 0.5s | 1m 47.4s | 7.5s |
| 3 | 0.5 | 7m 5.4s | 36 | 4.9s | 15.1s | 11.8s |
| 3 | 0.8 | 12m 13.0s | 37 | 7.2s | 25.2s | 19.8s |
| 3 | 1.0 | 16m 45.9s | 37 | 10.0s | 34.7s | 27.2s |
| 3 | 2.0 | 1h 23m 43.8s | 39 | 16.4s | 3m 47.9s | 2m 8.8s |
| 3 | 3.0 | 3h 12m 59.8s | 38 | 40.5s | 13m 59.1s | 5m 4.7s |
| 3 | 4.0 | 6h 42m 59.2s | 37 | 1m 0.4s | 3h 15m 39.2s | 10m 53.5s |
| 4 | 0.5 | 9h 4m 3.1s | 52 | 1m 8.0s | 15m 58.1s | 10m 27.7s |
| 4 | 0.8 | 1d 8h 18m 2.8s | 53 | 4m 33.3s | 58m 40.4s | 36m 34.0s |
| 4 | 1.0 | 3d 1h 12m 4.3s | 54 | 11m 37.2s | 2h 14m 38.6s | 1h 21m 20.1s |

**Table 3. Computation times for graph data.**

| n | $\lambda$ | Time | #Opt | Fastest | Slowest | Average |
|---|---|---|---|---|---|---|
| 2 | 0.5 | 1.3s | 24 | 0.0s | 0.0s | 0.0s |
| 2 | 0.8 | 2.2s | 22 | 0.1s | 0.1s | 0.1s |
| 2 | 1.0 | 3.1s | 20 | 0.1s | 0.2s | 0.1s |
| 2 | 2.0 | 1m 3.8s | 85 | 0.4s | 1.0s | 0.7s |
| 2 | 3.0 | 7m 42.4s | 146 | 0.7s | 6.5s | 3.2s |
| 2 | 4.0 | 55m 52.9s | 177 | 0.9s | 1m 1.5s | 18.9s |
| 2 | 5.0 | 7h 13m 55.2s | 190 | 0.9s | 6m 38.9s | 2m 17.0s |
| 3 | 0.5 | 8m 57.4s | 44 | 9.8s | 15.6s | 12.2s |
| 3 | 0.8 | 17m 15.5s | 48 | 17.3s | 25.1s | 21.6s |
| 3 | 1.0 | 36m 28.4s | 70 | 21.7s | 37.7s | 31.2s |
| 3 | 2.0 | 5h 29m 41.3s | 121 | 52.6s | 5m 32.5s | 2m 43.5s |
| 3 | 3.0 | 3d 4h 1m 24.7s | 167 | 58.3s | 1h 36m 20.6s | 27m 18.8s |
| 3 | 4.0 | 78d approx | 191 | 1m 9.9s | 1d 17h 42m 33.2s | 8h 20m 51.2s |
| 4 | 0.5 | 16h 19m 5.0s | 84 | 6m 6.6s | 16m 3.4s | 11m 39.2s |
| 4 | 0.8 | 2d 6h 37m 18.8s | 78 | 24m 59.9s | 54m 25.4s | 42m 0.9s |
| 4 | 1.0 | 5d 11h 35m 29.1s | 90 | 40m 33.8s | 2h 9m 50.7s | 1h 27m 43.6s |

values of $p$ need to be optimized to meet the requirements enforced in the plotting of the graphs (see above), resulting in a two-fold slow down where the amount of time per optimization increases as does the number of optimizations performed. We see this effect in the drastic increase in the time taken as $\lambda$ increases for fixed $n$ compared with the moderate increase in the average optimization times.

In the case of $n = 3$ with $\lambda = 4.0$ the 300 hour upper computation limit, corresponding resumptions, and unavoidable partial re-computation yield only an approximate computation time. A total of six interruptions (and thus resumptions) occurred during the computation, with the (partially) recomputed optimizations taking approximately 9, 15, 25, 32, 37, and 41 hours, respectively. As such, the reported time is an overestimate of the "true" computation time, and the discrepancy could be—in the worst case—in the vicinity of six days and fifteen hours. Nonetheless, even if the discrepancy is that large, it is not sufficient to undermine the broad pattern of growth we see as $\lambda$ increases.

## Conclusion

Determining an optimal experimental design for a growing population governed by a `POPBP` is a difficult problem. In this article, we developed a new approach to compute the Fisher information for higher values of $n$ and $\lambda$. With the use of generating functions, we constructed recursive equations for the likelihood function $\mathcal{L}_{Y_n}$ and its derivative, which we used to calculate the Fisher information. This approach allowed us to calculate the Fisher information more efficiently and accordingly determine which observation times maximize the Fisher information for given values of $n$, $\lambda$, and $p$.

For future work, we plan to develop further theoretical results on an optimal experimental design of a `POPBP` with the help of numerical experiments obtained in this paper.

We expect we can speed up the optimization process by using the drop values to rule out boundaries to check, thus computing fewer optimizations for any given combination of $n$,$p$, and $\lambda$. For example, for a fixed $n$ and $\lambda$, if we know we are optimizing for a value of $p$ that is larger than the upper bound of $\mathfrak{D}_1(\lambda)$, then we know $t_i^*$ is strictly less than 1, so we can avoid optimizing over any boundaries wherein $t_1 = 1$. This technique can even work with the binary search to find the drop values if we recall that our initial assumption was that $0 < \mathfrak{D}_i(\lambda) < 1$ for all $i$; however, we note that the implementation must be careful about how it handles cases where $p$ is *inside* multiple bounds simultaneously. An early test of this idea almost doubled the computation speed when we used the drop-value information (compared with the current method), but we have not yet finished implementation and testing, nor do we have properly rigorous timing data.

Finally, recent advances in GPU technologies (particularly in memory sizes) open the possibility of implementing the highly parallel nature of our computations on GPU hardware.

## Acknowledgments

We would like to thank Prof. Nigel Bean and Prof. Joshua Ross from University of Adelaide for discussing topics in Partially observable pure birth process, in particular the proof of Definition 5 with the first author. An early version of this work was discussed with Alin Bostan and Frédéric Chyzak at Inria in 2014.

## Author contributions

**Conceptualization:** Ali Eshragh, Bruno Salvy, Thomas McCallum.

**Formal analysis:** Ali Eshragh, Matthew Paul Skerritt, Thomas McCallum.

**Investigation:** Ali Eshragh, Matthew Paul Skerritt, Thomas McCallum.

**Methodology:** Ali Eshragh, Matthew Paul Skerritt, Bruno Salvy.

**Project administration:** Ali Eshragh.

**Software:** Matthew Paul Skerritt, Bruno Salvy.

**Visualization:** Matthew Paul Skerritt.

**Writing – original draft:** Ali Eshragh, Matthew Paul Skerritt, Thomas McCallum.

**Writing – review & editing:** Ali Eshragh, Matthew Paul Skerritt, Bruno Salvy.

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
