## [Decision Letter · Decision Letter 0]

20 May 2025

PONE-D-24-59633Optimal experimental design for partially observable pure birth processesPLOS ONE

Dear Dr. Skerritt,

Thank you for submitting your manuscript to PLOS ONE. After careful consideration, we feel that it has merit but does not fully meet PLOS ONE’s publication criteria as it currently stands. Therefore, we invite you to submit a revised version of the manuscript that addresses the points raised during the review process.

We look forward to receiving your revised manuscript.

Kind regards,

Hoda Bidkhori

Academic Editor

PLOS ONE

Journal Requirements:

Additional Editor Comments:

This paper presents a novel algorithm designed to identify optimal observation times by maximizing the Fisher information associated with the birth rate in a partially observable pure birth process involving nnn observations. In this context, "partially observable" signifies that, at each scheduled observation time, individuals in the process are independently observed with a fixed probability ppp, effectively modeling real-world challenges such as detection uncertainty or limited observational resources.

To address the computational complexity of this problem, the authors employ techniques from generating function theory, integrating symbolic and numerical computations to construct a recursive formulation for calculating the Fisher information. Although the recursion remains computationally intensive, it offers substantial improvements over previously existing methods, which proved to be computationally prohibitive even in the relatively simple cases. The paper demonstrates the efficacy of this approach through numerical experiments and provides a publicly accessible implementation.

I think this paper addresses an important problem in the design of experiments for partially observable stochastic processes. The authors introduce a novel approach by incorporating generating functions to compute and optimize the Fisher information, offering improvement over existing methods in terms of scalability and computational feasibility.

The manuscript has been evaluated by two reviewers—one recommending acceptance and the other suggesting minor revision. In light of these assessments, I recommend minor revision. The paper is methodologically sound and contributes meaningfully to the literature; however, I encourage the authors to address all comments and suggestions raised by both reviewers thoroughly and carefully to further strengthen the clarity and impact of their work.

Reviewers' comments:

Reviewer's Responses to Questions

**Comments to the Author**

1. Is the manuscript technically sound, and do the data support the conclusions?

Reviewer #1: Yes

Reviewer #2: Yes

2. Has the statistical analysis been performed appropriately and rigorously? 

Reviewer #1: Yes

Reviewer #2: Yes

3. Have the authors made all data underlying the findings in their manuscript fully available?

Reviewer #1: Yes

Reviewer #2: Yes

4. Is the manuscript presented in an intelligible fashion and written in standard English?

Reviewer #1: Yes

Reviewer #2: Yes

5. Review Comments to the Author

Reviewer #1: Title :Optimal experimental design for partially observable pure

birth processes .

After reviewing the research and its findings, I find that the study is well-conducted and its results are accurate and valid. Therefore, I recommend its acceptance in its current form.

Reviewer #2: This work addresses the optimal experimental design for partially observable pure birth processes (POPBPs), aiming to maximize the Fisher information with respect to the birth rate parameter. The authors introduce an innovative computational approach that leverages generating functions and symbolic computation to efficiently compute the Fisher information for these complex stochastic processes. This study is clearly relevant to researchers in stochastic modeling, applied probability, and statistical inference.

This work addresses the optimal experimental design for partially observable pure birth processes (POPBPs), aiming to maximize the Fisher information with respect to the birth rate parameter. The authors introduce an innovative computational approach that leverages generating functions and symbolic computation to efficiently compute the Fisher information for these complex stochastic processes. This study is clearly relevant to researchers in stochastic modeling, applied probability, and statistical inference.

The application of generating function techniques to optimize Fisher information in partially observable birth processes is novel and represents

a significant advancement over previous methods that were restricted to small values of $n$. The recursive formulation provides a practical computational tool, enabling computations for larger values of $n$ that were previously intractable. The theoretical derivations are well-structured, with clear formulations of generating functions, recurrence relations, and their implications for the likelihood and Fisher information.

The provision of a publicly available C++ implementation and GitHub repository further enhances the reproducibility and potential impact of the work.

In conclusion, this work is novel, interesting, and meaningful. Moreover, the presentation of this paper is clear and the proofs and computations appear to be correct. Therefore, I recommend that the paper be accepted for publication in $\textit{PLOS ONE}$.

However, the authors should carefully proofread the manuscript, as there are still some typographical and grammatical errors. For example:

\begin{enumerate}

\item The notation involving multiple indices (e.g., $\overline{y}_n$, $\overline{c}$, $q_{\overline{c}}$ ) may be overwhelming for readers unfamiliar with combinatorial generating functions. Consider adding a summary table of notation for reference;

\item Figure references (e.g., Fig. 1, Fig. 11) should be cross-checked to ensure correct linkage and appropriate caption placement;

\item Page 21, line 711: ``of of six days" $\longrightarrow$ ``of six days";

$\cdots$

\end{enumerate}

6. PLOS authors have the option to publish the peer review history of their article (what does this mean?). If published, this will include your full peer review and any attached files.

Reviewer #1: No

Reviewer #2: No

---

## [Author Response · Author response to Decision Letter 1]

6 Jun 2025

We sincerely thank the Editor and the two reviewers for their thoughtful and constructive feedback. We are grateful for their positive assessment of our work. We have carefully addressed all the comments raised, and we believe the revisions have improved the clarity and presentation of the manuscript. For ease of reference, the changes made in response to the comments have been highlighted in blue in the revised version. Below, we provide a detailed response outlining how each of the points in the referee report has been addressed.

Responses to Editor

-------------------

We thank you so much for considering our work and taking the time to read it carefully.

We have outlined below how we have fully addressed all the comments.

Responses to Reviewer #1

We thank you very much for your positive evaluation of our work and your recommenda- tion for acceptance. We really appreciate your encouraging feedback.

Responses to Reviewer #2

We thank you very much for your positive evaluation of our work and your recommendation for minor revision. Your encouraging feedback and constructive suggestions were greatly appreciated and have been instrumental in helping us enhance the clarity and quality of the manuscript. Below, we provide a detailed explanation of how each of your comments has been addressed.

1. The authors should carefully proofread the manuscript, as there are still some typographical and grammatical errors.

We have carefully proofread the entire manuscript and identified a few minor typographical errors, which are listed below. These, along with the items kindly noted by you, have all been corrected and are highlighted in the revised version.

(i) “optimisation” → “optimization” in the caption of Figure 2.

(ii) “grey” → “gray” on lines 565 and 566 of page 16.

2. The notation involving multiple indices (e.g., yn, c, qc ) may be overwhelming for readers unfamiliar with combinatorial generating functions. Consider adding a summary table of notation for reference.

We have added a new Table 1 at the end of the Introduction section to introduce all notation used throughout the paper. We hope this improves clarity and readability for the reader.

3. Figure references (e.g., Fig. 1, Fig. 11) should be cross-checked to ensure correct linkage and appropriate caption placement.

We have carefully reviewed all figure references and captions throughout the manuscript to ensure consistency and correct placement. All figures are appro- priately referenced, and their captions match the correct content. While we have ensured that captions appear immediately after the first mention of each figure, LaTeX may occasionally place a figure or its caption at the top of the following page (e.g., Figure 11). We have done our best to minimize such instances while adhering to journal formatting conventions.

4. Page 21, line 711: “of of six days” −→ “of six days”.

We have removed the redundant “of”.

---

## [Editor Report · Decision Letter 1]

6 Jul 2025

Optimal experimental design for partially observable pure birth processes

PONE-D-24-59633R1

Dear Dr. Skerritt,

We’re pleased to inform you that your manuscript has been judged scientifically suitable for publication and will be formally accepted for publication once it meets all outstanding technical requirements.

Kind regards,

Hoda Bidkhori

Academic Editor

PLOS ONE

Additional Editor Comments (optional):

Thanks for addressing the comments.
---

## [Editor Report · Acceptance letter]

PONE-D-24-59633R1

PLOS ONE

Dear Dr. Skerritt,

I'm pleased to inform you that your manuscript has been deemed suitable for publication in PLOS ONE. Congratulations! Your manuscript is now being handed over to our production team.

Kind regards,

on behalf of

Dr. Hoda Bidkhori

Academic Editor

PLOS ONE